



# swNEMO_v4.0: an ocean model NEMO4 for the next generation Sunway supercomputer

Yuejin Ye[1], Zhenya Song[2,3,4], Shengchang Zhou[2,4,5], Yao Liu[6], Qi Shu[2,3,4], Bingzhuo Wang[1], Weiguo Liu[4,5], Fangli Qiao[2,3,4], and Lanning Wang[4,7]

[1]National Supercomputing Center in Wuxi, 214000, China

[2]First Institute of Oceanography, and Key Laboratory of Marine Science and Numerical Modeling, Ministry of Natural Resources, Qingdao 266061, China

[3]Shandong Key Laboratory of Marine Science and Numerical Modeling, Qingdao 266061, China

[4]Laboratory for Regional Oceanography and Numerical Modeling, Pilot National Laboratory for Marine Science and Technology, Qingdao 266237, China

[5]Shandong University, Jinan 250101, China

[6]East China Normal University, Shanghai 200062, China

[7]Beijing Normal University, Beijing 100875, China

**Correspondence:** Fangli Qiao (email: qiaofl@fio.org.cn), Lanning Wang (email: wangln@bnu.edu.cn)

**Abstract.** The current large-scale parallel barrier of ocean general circulation models (OGCMs) makes it difficult to meet the computing demand of high resolution. Fully considering both the computational characteristics of OGCMs and the heterogeneous many-core architecture of the new Sunway supercomputer, swNEMO_v4.0 based on NEMO4, with ultrahigh scalability is developed. Three innovations and breakthroughs are shown in our work: (1) A highly adaptive, efficient four-level paral-

lelization framework for OGCMs is proposed to release a new level of parallelism along the compute-dependency column dimension. (2) A many-core optimization method using blocking by remote memory access (RMA) and a dynamic cache scheduling strategy, effectively utilizing the temporal and spatial locality of data. The test shows that the actual DMA bandwidth is greater than 90% of the ideal bandwidth after optimization, and the maximum is up to 95%. (3) A mixed-precision optimization method with half-, single-, and double-precision is explored, which can effectively improve the computation per-

formance, while maintaining the simulated accuracy of OGCMs. The results demonstrate that swNEMO_v4.0 has ultrahigh scalability, achieving up to 99.29% parallel efficiency with a resolution of 500 m using 27,988,480 cores, reaching the peak performance with 1.97 PFlops.

## 1 Introduction

Ocean general circulation models (OGCMs) are numerical models focusing on the properties of oceans based on the Navier-

Stokes equations on the rotating sphere with thermodynamic terms for various energy sources (Chassignet et al., 2019). OGCMs are the most powerful tools for predicting the ocean and the climate states. As shown in Figure 1, recent studies indicate that the horizontal resolution of OGCMs used for ocean research has been improved from 5 degrees (approximately 500 km) (Bryan et al., 1967; Bryan, 1969) to 1/48 degrees (approximately 2 km) (Rocha et al., 2016; Viglione et al., 2018;



Dong et al., 2020; Qiu et al., 2018, 2020). However, the small-scale processes in the ocean (Chassignet et al., 2019; Lellouche

et al., 2018), which are critical for further reducing the ocean simulation and prediction biases, still cannot be resolved within

a 2-km resolution. Therefore, improving the spatial resolution is the one of most important directions of OGM development.

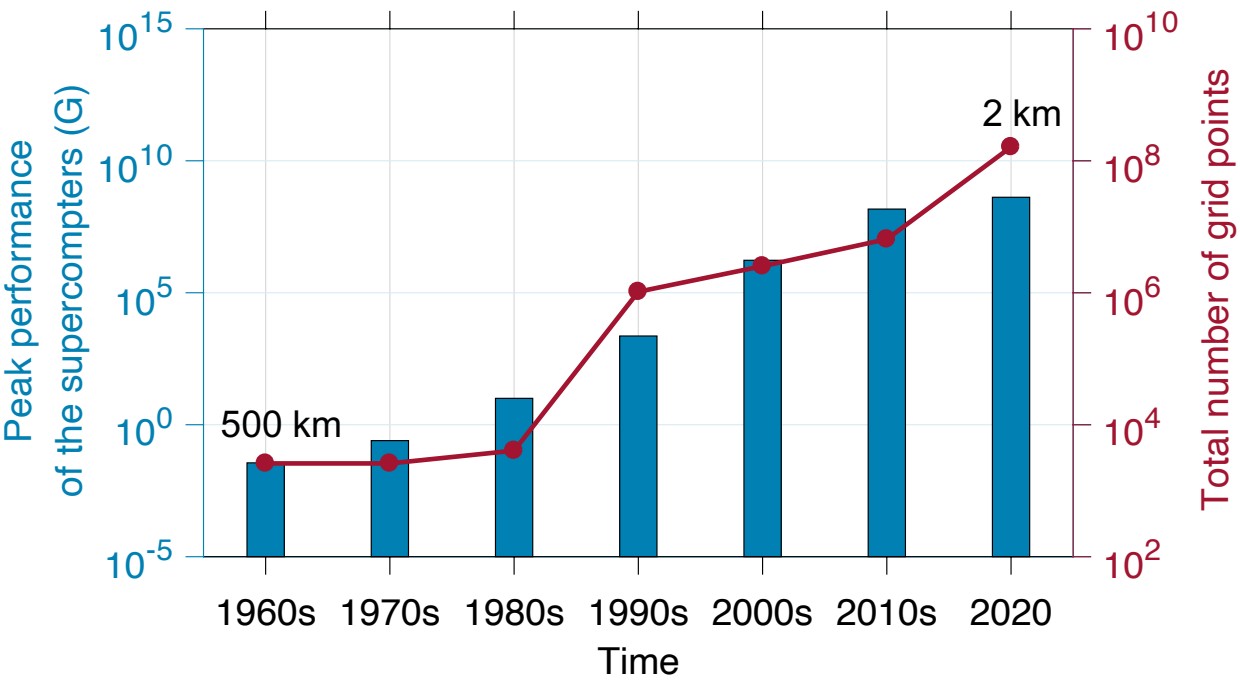

**Figure 1.** Peak performance of the supercomputers (blue bar) and the total number of OGCM grid points (red line) over the past 60 years.

The increase in OGCMs resolution results in an exponential increase in the demand for computing capabilities. With the

horizontal resolution doubled, the amount of calculation increases about 10 times accordingly. The enhanced performance

of supercomputers, however, makes it feasible to simulate oceans at higher resolutions. Compared to homogeneous systems,

which cannot afford the high-power cost of the transition from PetaFlops supercomputers to ExaFlops supercomputers, hetero-

geneous many-core systems reduce the power loss from the perspective of system design, thus, becoming mainstream.

In the last few years, heterogeneous architectures (*i.e*, CPU+GPU (Vazhkudai et al., 2018), CPU+FPGA (Putnam et al.,

2014), CPU+MIC (Liao et al., 2014) and MPE+CPE (Fu et al., 2016)) have been widely applied to speed up model computation

(Table 1). GPUs play an increasingly important role in high-performance computing due to their high compute density. Using

GPUs to accelerate computing speed shows great potential for ocean and climate modeling. For example, based on mpiPOM,

Xu et al. developed a POM GPU solution, which reduced power consumption by 6.8×, and achieved the performance of 408

Intel Westmere cores on four K20 GPUs (Xu et al., 2015). The performance of FUNWAVE-GPU developed by Yuan et al. on

2 v100 GPUs was 12× on that of 36 cores (2 Intel Xeon Gold 6150) (Yuan et al., 2020). LICOM was successfully ported and



**Table 1.** Research on models based on heterogeneous architectures

| Year | Models | Maximum Resolutions | Platforms | Maximum Scales | Parallel Programming | Results |
|------|--------|---------------------|-----------|----------------|----------------------|---------|
| 2012 | Hydrostatic LAM | 13,600×3,333×30 | Intel Xeon E5506 | 12 | MPI+FPGA | 74× speedup |
| 2015 | POM.gpu | 1,922×1,442×51 | NVIDIA TESLA K20X | 4 | MPI+CUDA | Equivalent to 408 Westmere cores performance |
| 2016 | MUSNUM | 387,911,175 | TaihuLight | 10,649,600 cores | MPI+Athread | 36.22% efficiency, 45.43 PFlops |
| 2019 | LICOM2 | 360×218×30 | NVIDIA Tesla K80 | 4 | MPI+OpenACC | 6.6× the 4 Intel Xeon CPU E5-2690 v2 GPUs. |
| 2020 | FUNWAVE-GPU | 9,600×7,200 | NVIDIA V100 | 2 | MPI+CUDA | 12 times the performance of 36 cores (Intel Xeon Gold 6150) |
| 2020 | LICOM3 | 7,200×3,920×55 | AMD GFX906 GPU | 26,200 | MPI+HIP | 2.72 SYPD |
| 2020 | POP2 | 3,600×2,400×65 | TaihuLight | 1,189,500 cores | MPI+Athread | 3.8× speedup |
| 2020 | CESM-HR | 3,600×2,400 | TaihuLight | 4,004,000 cores | MPI+Athread | 3.4× speedup |
| **2022** | **swNEMO_v4.0** | **82,500×55,000×128** | **Next Generation Sunway** | **27,988,480 cores** | **MPI+Athread** | **∼8× speedup, 99.29% parallel efficiency** |

optimized to run on Nvidia and AMD GPUs, which showed great potential compared with CPUs (Jiang et al., 2019; Wang et
al., 2020). Compared with GPUs, the results of ocean models published with FPGA are relatively few, and most of them are in
the program porting stage.

On the platform of Sunway supercomputers, several researchers (Zhang et al., 2020) and (Zeng et al., 2020) studied the port-
ing of CESM and its ocean component, POP2, and successfully scaled to 1 million and 4 million cores, respectively, speeding
up computation by 3-4×. An irregular parallel decomposition scheme was applied to China's self-developed MASNUM ocean
service wave model and scaled to tens of millions of cores (Qiao et al., 2016). To use heterogeneous architectures, parallel
programming of MPI + OpenMP/CUDA/OpenACC/Athread was developed to improve the parallel efficiency of the model us-
ing finer-grained parallelism (Afzal et al., 2017). The coordination between the master and slave cores (involving master-slave
cores, slave-slave cores, memory bandwidth and register communication) is the key to parallel efficiency.

Due to the limitation of memory bandwidth, large-scale parallelism usually cannot maintain high efficiency (Lellouche et al.,
2018; Ruston, 2019; Chassignet et al., 2019; Lellouche et al., 2018). For example, with CAM-SE ported to TaihuLight, the per-
formance only improved by approximately 2× with 24,000 core groups (CGs) with a resolution of 25 km (Fu et al., 2016). The
classic parallelization method is the "longitude-latitude" 2D decomposition based on the MPI of OGCMs. To further improve
the parallelism, there are many efforts in parallel decomposition schemes and algorithm improvements of physical processes.
To achieve better load balance, several decomposition schemes based on curve filling have been introduced in OGCMs (POP
(Smith et al., 2010) and NEMO (Madec et al., 2016)) and in the MASNUM (Zhao et al., 2014) ocean wave model. The parallel
scale can reach $O(100km)$ in practice with a resolution of $O(10km)$ (Hu et al., 2013; Yang et al., 2021). However, due to the
communication barrier, the large-scale parallel efficiency is still below 50%. Therefore, we need to explore new schemes that
can further improve the scalability to the next level. At the current stage, the development of a large-scale parallel algorithm
of the OGCMs is still in two-dimensional parallelism along the longitudinal and latitudinal horizontal directions. Therefore,
integrating the vertical direction in the three-dimensional parallelism scheme is still a challenging problem.



As most emerging Exa-scale systems provide support for mixed-precision arithmetic, a mixed-precision computing scheme becomes an important step to further reduce the computational and memory pressure, as well as to improve the computing performance. However, due to the weak support of mixed precision from computer systems and the difficulty of balancing computing precision and results accuracy, many efforts are still at an early stage. Dawson *et al.* used a reduced-precision emulator (RPE) to study the shallow water equation (SWE) model using mixed precision and found that the error caused by iterations with low precision can be solved by mixed precision (Dawson et al., 2017). Then, based on the RPE, Prims *et al.* investigated the application of a mixed scheme of double precision (DP) and single precision (SP) on NEMO and verified the feasibility of half precision (HP) in the regional ocean model of ROMS (Prims et al., 2019). Previous studies have shown that mixed precision can improve computational efficiency. However, RPE can only verify the feasibility of mixed precision in theoretical models. The new generation Sunway supercomputer, supporting HP, can lay a solid foundation for applying mixed-precision OGCM.

The architecture of the new generation of Sunway processors (SW26010 Pro) adopts more advanced DDR4 compared with original SW26010. It not only expands the capacity but also greatly improves the DMA bandwidth of the processor. The upgrade of the on-chip communication mechanism of many-core arrays makes the interconnection between CPEs more convenient and builds a more efficient global network on the entire supercomputer. The upgrade of these key technologies provides a solid foundation for the parallelization of OGCMs on the new generation of Sunway supercomputers. In this work, we design and implement a four-level parallel algorithm on three-dimensional space based on hardware-software codesign. We also resolve the problem of memory bandwidth through fine-grained data reuse technology, thus paving the way for the ultrahigh scalability of NEMO. Furthermore, based on Sunway's heterogeneous many-core architecture, a composite block algorithm and a dynamic scheduling algorithm based on LDCache are proposed to fully exploit the performance of many-core acceleration. Finally, half precision is introduced to further release the memory pressure of NEMO under simulation with ultrahigh resolution. We develop a highly scalable swNEMO_v4.0 based on the Nucleus for European Modelling of the Ocean version 4 (NEMO4) with the GYRE-PISCES benchmark (Madec et al., 2016). The resolution is equivalent to the horizontal resolution of 500 m on a global scale.

In the following section, we briefly introduce the new generation of Sunway heterogeneous many-core supercomputing platforms. In Section 3, we briefly review the basics of NEMO4 and describe our optimization methods in detail. The performance results are discussed in Section 4, and Section 5 concludes this paper with discussions.

## 2  The New Generation Heterogeneous Many-core Supercomputing Platform and NEMO4

Succeeded by the architecture of Sunway TaihuLight, the new Sunway generation, as shown in Figure 2, which is driven by SW26010 Pro, consists of 6 core groups, with one management processing element (MPE) and one $8 \times 8$ computing processing element (CPE) cluster in each. The core groups are connected within loop network. Data can be transferred between CPEs via remote memory access (RMA), which significantly increases the efficiency of CPE co-working.





MPE is based on SW64 instruction set, with a 32KB L1 instruction cache, a 32KB L1 data cache and 512KB L2 cache. CPE is also based on SW64 instruction set, with the 512-bit single-instruction multiple-data (SIMD) vector, where double precision, single precision, half precision, and integer are all supported. Furthermore, double precision and single precision share the same computation speed whereas half precision performs twice faster. There are also a separate instruction cache and a scratchpad memory (SPM) in each CPE. The SPM allocates local data memory (LDM) for users, part of which can be set as local data cache, automatically administrated by hardwares. The data is transmitted between LDM and main memory via either direct memory access (DMA) or the general load and save instructions.

**Table 2.** Basic information about SW26010 Pro

| CPU | SW26010 Pro |
| --- | --- |
| Number of CGs | 6 |
| Number of processors of one CG | 65 (1 MPE + 64 CPE) |
| Programming language | C/C++, Fortran, Python |
| Parallel Programming Environment | MPI, Athread/OpenACC |
| Memory Size | 96GB |
| Instruction Set | sw64 |
| L1 Instruction Cache | 32KB |
| L1 Data Cache | 32KB |
| L2 Data Cache of MPE | 512KB |
| SIMD register of MPE | 256-bit |
| SIMD register of CPE | 512-bit |
| Precision Support | Double-precision |
| | Single-precision |
| | Half-precision |
| Scratch Pad Memory | 256KB |

NEMO is a state-of-the-art modeling framework for research activities and forecasting services in ocean and climate sciences developed in a sustainable way by a European consortium since 2008. It has been widely used in marine science, climate change studies, ocean forecasting systems and climate models. For ocean forecasting, NEMO has been applied by many operational forecasting systems, such as the Mercator Ocean monitoring and forecasting systems (Lellouche et al., 2018), as well as the ocean forecasting system in the National Marine Environmental Forecasting Center of China with a 1/12 degree resolution (Wan, 2020). For the climate simulation and projections, approximately 1/3 of the climate models in the latest phase of Coupled Model Intercomparison Project Phase 6 (CMIP6) (Eyring et al., 2016) use the ocean component models of NEMO. The breakthroughs made in this study are based on NEMO, and thus, this work provides useful methods and ideas that can directly contribute to the NEMO community for optimizing their models and improving their simulation speeds.



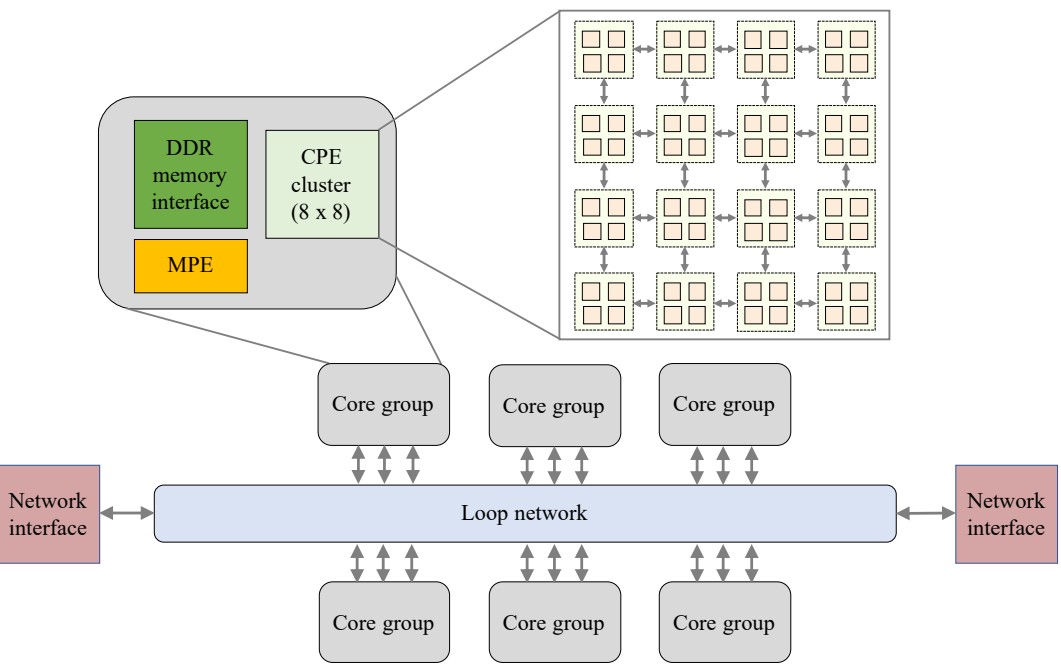

**Figure 2.** Architecture of the new generation Sunway supercomputer.

## 3    Porting and Optimizing NEMO4

Fully considering the characteristics of the three-dimensional spatial computation of OGCMs and the heterogeneous many-core architecture of the new generation Sunway supercomputer, we develop a highly scalable NEMO4 named swNEMO_v4.0, with the following three major contributions based on the concepts of hardware-software co-design:

– A highly efficient four-level parallelization framework is proposed for OGCMs to release a new level of parallelism along the column dimension that was originally not parallel-friendly due to the computational dependency;

– A raised many-core optimization method that uses an effective dynamic cache scheduling strategy to effectively utilize the temporal and spatial locality of data;

– A multi-level mixed-precision optimization method that uses half-, single-, and double-precision is explored, which can effectively improve the computation performance while maintaining the same level of accuracy;

We then elaborate on the above algorithms in the following.

### 3.1    An Adaptive Four-level Parallelization Framework

To utilize the many heterogeneous cores of the new generation Sunway supercomputer, the load of the expanded C-grid computation should be assigned in a balanced way. Considering the characteristics of the three-dimensional spatial computation



of NEMO and the heterogeneous many-core architecture of a new generation Sunway supercomputer, we propose an adaptive four-level parallel framework that can realize computation load dispatching among different levels. Figure 3 demonstrates our

adaptive four-level parallel framework.

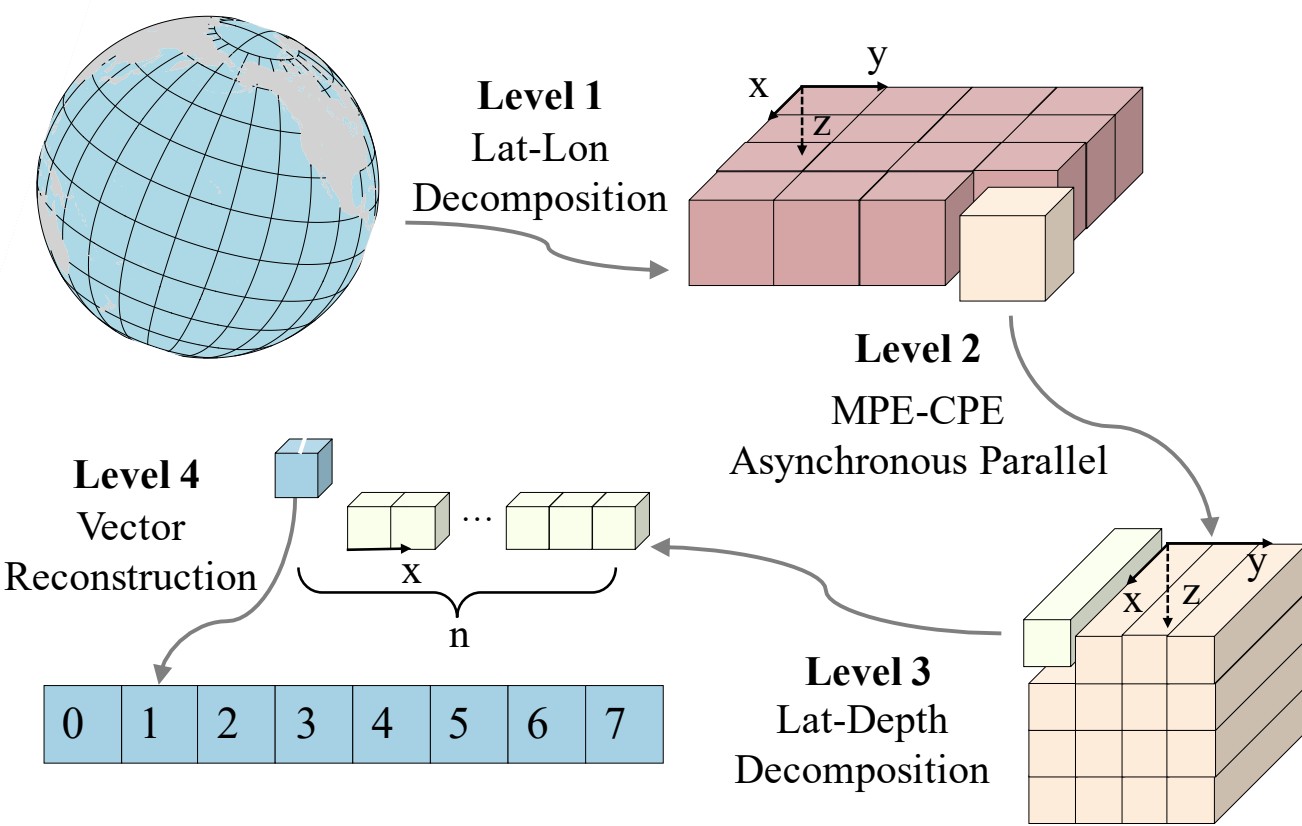

**Figure 3.** The adaptive four-level parallel framework, where $x$-, $y$-, and $z$-axes indicate latitude, longitude and depth, respectively. Level 1 is the load-balanced "longitude-latitude" decomposition among MPEs, level 2 is the asynchronous parallel between a MPE and a CPE cluster, level 3 is the "Latitude-depth" decomposition in a CPE clusters and level 4 is the vector reconstruction in a CPE.

### 3.1.1    Load-balanced "Longitude-Latitude" Decomposition among MPEs

With land grids eliminated in NEMO, keeping the horizontal data in a regular way benefits the load balance among processes. The variables are stored as 3-D arrays, and the $x$, $y$ and $z$ axes represent the longitude, latitude and depth, respectively. Since the scales of the $x$-axis and $y$-axis are considerably larger than that of the $z$-axis, we decompose the data along the $x$-axis and

$y$-axis and dispatch each data block to different MPI processes. Our major strategy is to guarantee that (a) the grid sizes of the subdomain in each process are similar, achieving a good load balance; and (b) the $x$-axis and $y$-axis dimension sizes are close





to each other. The *x*-axis and *y*-axis of each process are the closest in all options, thus minimizing the halo areas that require communication among processes.

### 3.1.2 Asynchronous Parallelization between a MPE and a CPE Cluster

Considering that MPE is asynchronous with a CPE cluster, we propose an asynchronous parallelization design between the MPE and the CPE cluster by using efficient DMA. The MPE is in charge of the boundary data exchange, I/O and a small amount of computation, while the CPE cluster performs the computation of most kernels. Such an asynchronous communication pattern can make full use of the asynchronous parallelism between the MPE and the CPE cluster, thus improving the parallel efficiency.

### 3.1.3 "Latitude-depth" Decomposition in the CPE Cluster

Furthermore, we design the third level of parallelism in the CPE cluster by utilizing the fine-grained data-sharing features within the CPE cluster, releasing a new level of parallelism along the column dimension that was originally not parallel-friendly due to the computational dependency. RMA is a unique on-chip communication mechanism on the next-generation Sunway supercomputer, which enables communication among CPEs with high speed. We realize the "latitude-depth" decomposition with LDM and the data exchange with RMA. By utilizing the row and column communication features of RMA to achieve

fine-grained data sharing among the CPE threads, we can accomplish an efficient parallelization on the column (*i.e.*, the depth) dimension and release a new level of parallelism for the underlying hardware.

In the data structure of swNEMO_v4.0, data items along the longitude axis are stored in a continuous way. Following such a memory layout, we divide the data block along the latitude and depth into smaller blocks and copy them into the LDM. More details are shown in Section 3.2.

### 145 3.1.4 Data Layout Reconstruction for Vectorization

To further improve the computational efficiency within each CPE, we design the fourth level of parallelism for vectorization. SW26010 Pro has a 512-bit SIMD instruction set, one of which can compute 8 double numbers at a time. To adapt our computation pattern for the SIMD instruction, we perform a data layout reconstruction to achieve the most suitable vectorization arrangement. Moreover, methods, such as instruction rearrangement, branch prediction and cycle expansion are adopted to

improve the execution efficiency of the instruction pipeline.

## 3.2 Performance Optimization for the Many-core Architecture

### 3.2.1 A Composite Blocking Algorithm Based on RMA

To further improve the parallel scale and efficiency, we implement the fine-grained "lat-depth" decomposition. There are many stencil and temporal dependency computations existing in NEMO, which results in a high demand for DMA bandwidth. A

stencil computation is a class of algorithms that updates elements in a multidimensional grid based on neighboring values using a fixed pattern, here after called stencil. In a stencil operation, each point in a multidimensional grid is updated with





the weighted contributions from a subset of its neighbors in both time and space, thereby representing the coefficients of the partial differential equation (PDE) for that data element. Therefore, we take advantage of the RMA provided by SW26010 Pro to relieve the pressure of the DMA bandwidth. RMA is an on-chip communication mechanism with superior bisection
bandwidth within a CPE cluster. RMA enables direct remote LDM access among different CPEs within one CPE cluster. The efficient batch communication mechanism of RMA is highly adaptable for solving typical *x*-pointer problems. For example, in the diffusion process of the *tracer* in NEMO, upstream points are needed for data exchange, which includes horizontal unidirectional grid information exchange (3-pointer) and grid information exchange along longitude and latitude directions (5-pointer). Figure 4 represents the grid communication process between CPE #0 and CPE #1. Each point represents a multi-
dimensional tensor composed of different variables that are irrelevant to each other. When updating $A(u)$, variables $v$ and $w$ from the surrounding 8 points are required to participate in the calculation. We first send $v$ and $w$ required by adjacent CPEs to the buffer in the corresponding CPEs while updating the local variables whose surrounding points are on the same CPE. After all the needed variables in the halo are transferred into the buffer, the remaining variables in the points located at the edge of the CPE can be updated.

Based on RMA, we design different parallel blocking algorithms for computing kernels with different characteristics to maintain an efficient performance in various application scenarios.

- Temporal dependency in computing along the *z*-axis. We suppose $u_{i,j,k-1}^{iter}$ is an original value of grid $(i, j, k-1)$ in the coordinate system $(x, y, z)$ (where the *x*-axis is the most continuous one and the *z*-axis is the least) and $u_{i,j,k}^{iter}$ is computed as follows:

$$\begin{cases} u_{i,j,k}^{iter} = f(u_{i,j,k-1}^{iter}) \\ \quad f = \alpha_1 \times u_{i,j,k-1}^{iter} + \beta_1 \times (u_{i\pm1,j,k}^{iter} + u_{i,j\pm1,k}^{iter}) \end{cases}$$

where $0 < i < n_x$, $0 < j < n_y$, and $1 < k < n_k$. To fully utilize the DMA bandwidth, we decompose data along the *y*-axis and send them into different CPEs. Then, continuous data along the *x-axis* are copied into the LDM at a time.

- Temporal dependency in computation along the *y*-axis. In this case, the computation is as follows:

$$\begin{cases} u_{i,j,k}^{iter} = f(u_{i,j-1,k}^{iter}) \\ \quad f = \alpha_2 \times u_{i,j-1,k}^{iter} + \beta_2 \times (u_{i\pm1,j,k}^{iter} + u_{i,j,k\pm1}^{iter}) \end{cases}$$

Restricted by the size of the LDM, we decompose data along the *z*-axis and then decompose data along the *y*-axis into the proper size $m$ and dispatch each data block into different CPEs as follows:

$$m = \frac{\text{size(LDM)}}{\text{size}(x)}$$

Therefore, blocks on the same *z*-layer are dispatched to the same CPE one-by-one.

- Nontemporal dependency in computation. In this case, we decompose the data along the *z*-axis and *y*-axis. Due to the elaborate design of size $(x)$, each block can be copied into LDM at once to reduce redundant halo transfer and improve the asynchronous parallel of stencil calculations.



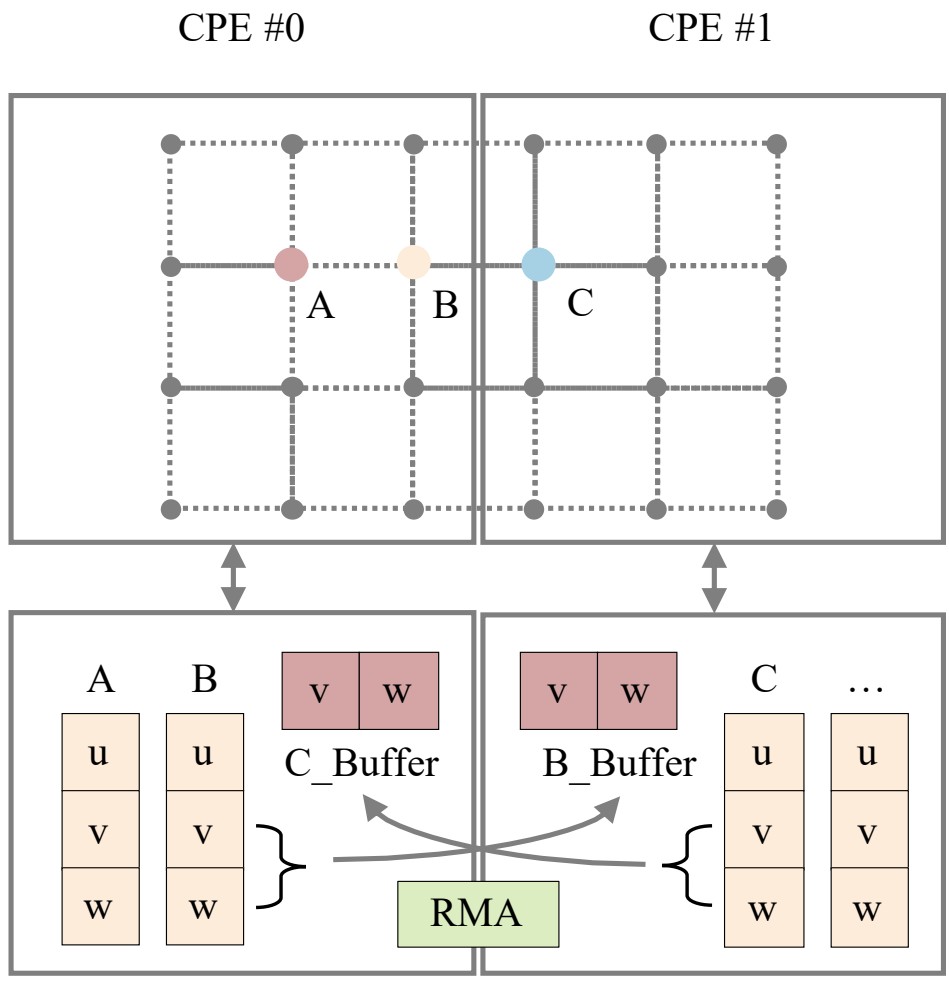

**Figure 4.** Tensor distribution in different CPEs (upper) and data transportation between CPEs (lower).

To increase the utility of the RMA bandwidth, we pack the data before sending for a better utilization of the bandwidth in an aggregated manner. While sending, CPEs compute, in this way, we can realize the overlap between RMA communication and computation, and thus, make efficient utilization of the bandwidth by reducing redundant DMA references. By utilizing RMA and the composite blocking strategy, for nontemporal dependency and temporal-dependency computation, we finally achieve over 90% effective utilization of the RMA bandwidth and over 80% utilization of the DDR4 memory bandwidth. With such an efficient memory scheme, the average speedup of most compute kernels (comparing the performance of a CPE cluster to an MPE) can be up to $40\times$.





### 3.2.2 A Dynamic LDCache Scheduling Algorithm

To access discrete data items in swNEMO_v4.0, the traditional global load/global store (gld/gst) method becomes a major performance hinderance. LDCache, provided by SW26010 Pro, stores data with a cache_line of 256ytes in CPE. In one CPE, LDM and LDCache share an SPM with a total capacity of 256 KB, and the size of LDM and LDCache can be manually adjusted. Since the amount of data required for a round of computation in different kernels in NEMO is varied, if the cache space is fixed, the utilization of the LDM becomes less efficient. Therefore, we design a dynamic LDCache scheduling algorithm that can realize efficient and fine-grained memory access. One feature of this algorithm is to dynamically adjust the size of LDCache to achieve a balance between LDM and LDcache. Furthermore, since LDCache cannot guarantee data consistency with memory, the algorithm has a time-division update technique. We regularly update the stored data and eliminate the outdated data in the LDM simultaneously. As shown in Figure 5, the data that needs to be refreshed is packed on the CPE and sent to the designated buffer of the MPE. The MPE then uses the MPE-CPE message mechanism to find the buffers that need to be updated in a round-robin way and update them, while the CPE eliminates the corresponding data in the cache. By applying the dynamic LDCache scheduling algorithm with both an adjustable cache and a manual time-division update technique, we can improve the memory bandwidth utilization rate to approximately 88.7% for DDR4, with an 88× speedup (comparing the performance of a CPE cluster to an MPE) for most computing kernels, which is a substantial improvement compared with the 5.1× speedup when using the traditional gld/gst method.

### 3.3 Mixed-precision Optimization

As NEMO, an OGCM, is memory intensive, the memory bandwidth limits the computational efficiency of NEMO to a large extent. A reduced-precision method can be a promising solution. However, using low-precision data is a double-edged sword. On the one hand, it can effectively resolve the performance obstacles ; on the other hand, the reduced precision brings errors and uncertainties. According to (Dawson et al., 2017), 95% of the variables in NEMO support the single precision floating-point format (SP). Therefore, we specifically reconstruct the data structure and introduce a new three-level mixed precision scheme in NEMO. To achieve a higher performance (Figure 6), we reconstruct two "half precision + single precision" (HP+SP) computing kernels *tracer_fct* and *tracer_iso* of NEMO on CPE, which account for 50% of the hotspot runtime in total. As HP is only supported on the CPEs of SW26010 Pro, we store HP data in the format of char or short types of memory on the MPE and use HP format to read them to LDM by DMA to ensure the correctness on the CPE.

By analyzing the calculation characteristics of NEMO, we find that the HP format subtracts operations between adjacent grids, which makes the final results diverge due to the rounding error of low precision. Therefore, a high-precision format must be used when calculating adjacent grids. Based on the above analysis in other calculations, we adopt the SP floating-point format for calculations between adjacent grids and the HP floating-point format with BF16 consists of 1 sign bit, 8 exponent bits and 7 mantissa bits. After optimization, we achieve a speedup close to 2× compared with NEMO using DP, while the maximum biases of temperature, salinity and velocity are still within 0.05% (figures not shown). Therefore, utilizing mixed

**Figure 5.** Time-division refresh technology for LDM.





precision can effectively increase the computational intensity and improve the scalability while maintaining the simulated accuracy of NEMO.

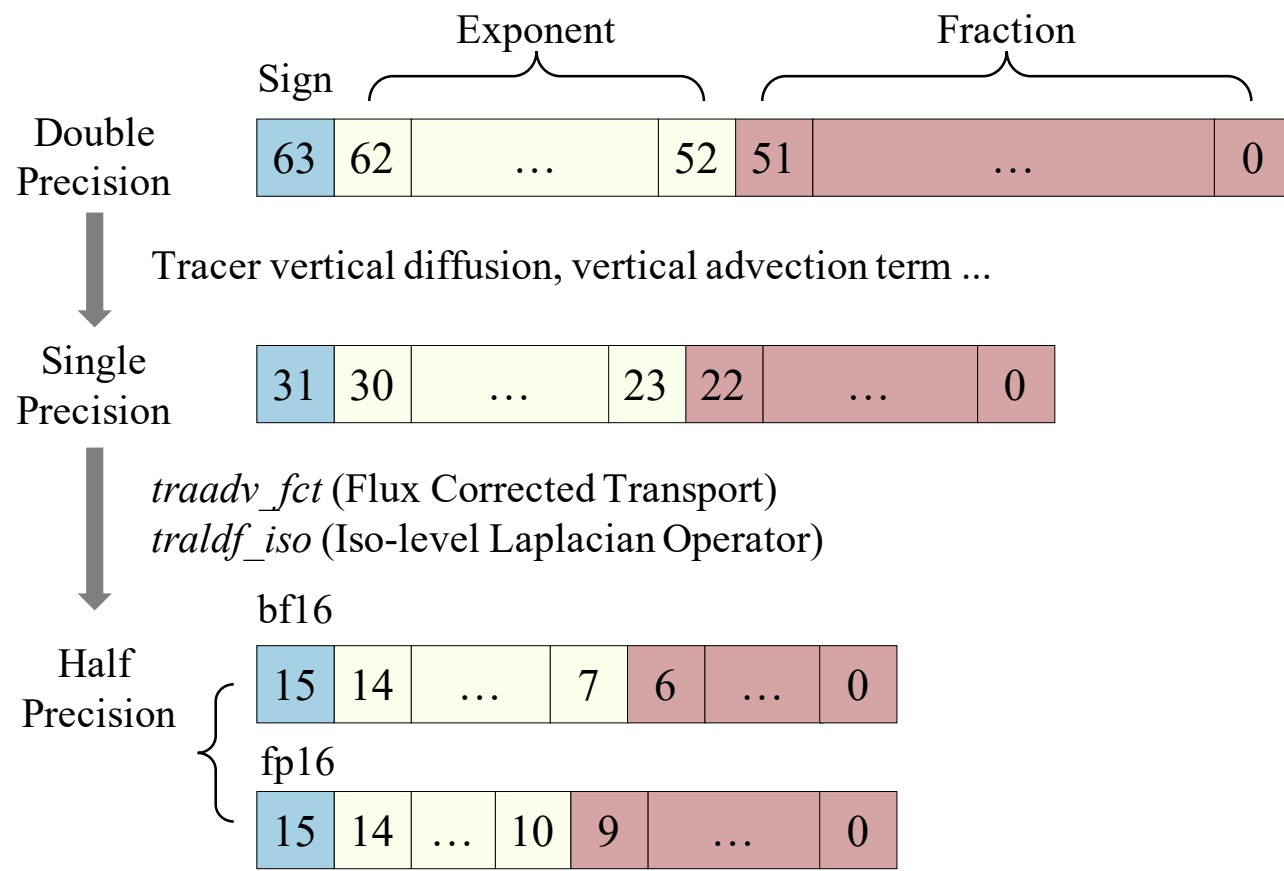

**Figure 6.** Data reconstruction for three-level mixed precision.

## 4 Performance Results

We choose the benchmark named GYRE-PISCES to test the swNEMO_v4.0 performance. The domain geometry is a closed rectangular basin on the $\beta$-plane centered at $\sim 30°$N and rotated by $45°$, 3,180 km long, 2,120 km wide and 4 km deep. The circulation is forced by analytical profiles of wind and buoyancy fluxes. This benchmark represents an idealized North Atlantic or North Pacific basin (Madec et al., 2016). In addition, the East-West periodical conditions and the North Pole folding of the global ocean with a tripolar grid have a large impact on performance. Therefore, we activate the BENCH option to include these periodicity conditions and reproduce the communication pattern of the global ocean with a tripolar grid between two North Pole subdomains. It is equivalent to a global ocean with a tripolar grid with the same number of grid points from the





perspective of computational cost and computational characteristics, although the physical meaning is limited. In the following content, the resolution of the benchmark is equivalent to that of the global ocean.

In this work, we design three experiments with 2 km, 1 km, and 500 m horizontal resolutions for the strong scalability
analysis. Each experiment consists of 8 different parallel scales (Table 3). The speedup is equal to the clock time in different scales divided by the baseline record of the minimum scale with 2,129,920 cores. For the weak scalability analysis, we design the experiment with 8 resolutions (Table 3). All experiments are run for 1 model day without I/O. We adopt the following methods to perform statistics on time and floating points to ensure the measurement accuracy.

- For time statistics, we use two methods to proofread:

- We use the MPI_Wtime() function provided by MPI to obtain the wall-clock time;

  - We use the assembly instructions to count the cycle time.

- Similarly, two methods are used in the statistics of floating-point operations:

  - We use the loader to count the floating-point operation of the program when submitting the job;

  - We use performance interface functions to perform the program instrumentation to count the operations.

**Table 3.** Eight different scales used in weak scalability, the conversion between resolution in *degree* and resolution in *km*, and the total number of computed grid points used in weak scalability.

| Scales in weak scalability (cores) | Resolution (°) | Resolution (km) | Computed grid points (horizontal) |
|---:|:---:|:---:|---:|
| 2,129,920 | 1/12 | 9.0 | 13,515,004 |
| 4,259,840 | 1/16 | 7.0 | 24,020,004 |
| 8,519,680 | 1/24 | 4.5 | 54,030,004 |
| 12,779,520 | 1/32 | 3.5 | 96,040,004 |
| 17,039,360 | 1/44 | 2.5 | 181,555,004 |
| 21,299,200 | 1/64 | 2.0 | 384,080,004 |
| 25,559,040 | 1/96 | 1.2 | 864,120,004 |
| 27,988,480 | 1/116 | 1.0 | 1,261,645,004 |

## 4.1 Parallel-Working Performance on a Many-Core Architecture

The cycle time and speedup ratio of the hotspots are shown in Figure 7, where different parallel methods are applied for comparison with the original method. While the CPEs parallel method, which introduces all 64 slave kernels to help with acceleration, is 12 times faster than the original method, we can surprisingly find that the master-slave asynchronous parallel-working mode (MPE-CPEs multilevel parallel) shows quite satisfying results with a 65× speedup. The four-level parallel framework fits well



in Sunway architecture, and the master-slave asynchronous parallel-working strategy combines data parallelism with task par-
allelism. By making great use of the existing Sunway architecture, this parallel-working strategy overlaps the computation part
with the communication and I/O part, thus considerably improving hotspot efficiency.

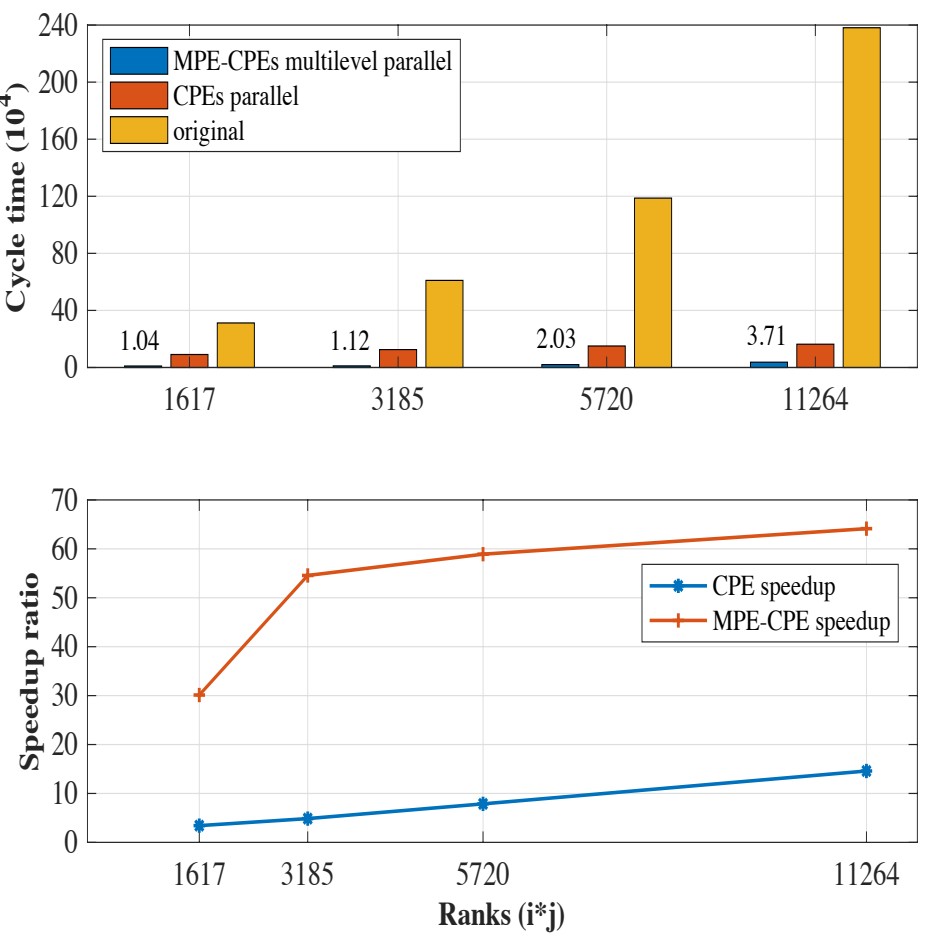

**Figure 7.** Performance of MPE-CPE asynchronous parallelization

The Sunway architecture suggests storing data into slave kernels before further computation. Therefore, the performance
of the many-core architecture is related to the actual amount of memory access, the bandwidth of DMA and the FLOPS
performance of slave kernels. We assume the total amount of time of the hotspot part in slave kernels is represented by
$T = T_c^{'} + T_l^{'}$, where $T_c^{'}$ is the time cost of the actual floating-point operation in this part, and $T_l^{'}$ is the time cost of direct
memory access. To speed up the many-core architecture, we can seek methods for both DMA optimization and floating-point
operation optimization.





For stencil computation, due to the low ratio of computation to memory access, we have $T_c^{'} < T_l^{'}$. We can minimize $T_c^{'}$ using computation-communication overlap, yet $T_l^{'}$ is related to the actual amount of memory access and the bandwidth of DMA $T_l^{'} \geq \frac{M}{BW_{DMA}^{'}}$, where $BW_{DMA}$ is the theoretical value of DMA bandwidth and $M$ is the total valid amount of memory access. In the real NEMO4 case, we have $T_c^{'} << T_l^{'}$. Assuming the following equation holds true, $BW_{DMA}^{'} = \frac{M}{T_c^{'}+T_l^{'}}$, then the ratio of actual DMA bandwidth to theoretical bandwidth should be

$$\alpha = \frac{\frac{M}{T_c^{'}+T_l^{'}}}{BW_{DMA}} = \frac{M}{(T_c^{'}+T_l^{'})BW_{DMA}}$$

The theoretical time cost of DMA when M is fixed is $T = \frac{M}{BW_{DMA}}$. Since $T_l^{'}$ is affected by the frequency and size of DMA, $T < T_l^{'}$ and $T < T_c^{'} + T_l^{'}$ hold. Therefore, we have

$$\alpha = \frac{M}{(T_c^{'}+T_l^{'})BW_{DMA}} = \frac{T}{T_c^{'}+T_l^{'}} < 1$$

when $T_c^{'} \to 0$ or $T_c^{'} \to T$, $\alpha \to 1$. The more the ratio of actual memory bandwidth to theoretical bandwidth approaches 1, the faster the whole architecture works, and thus, the better the performance becomes.

To analyze the performance of the algorithm mentioned in Section 3.2.1, we use five kernels to simulate the test, with grids 49×65×128 on average per kernel. We calculate the average clock cycle of running one simulation, and the results are shown in Figure 8(see the appendix for the detailed codes of all the following kernels 1∼5), where the red bars are the original results and the blue bars are the optimized results using this algorithm. It is clearly shown that the time cost is largely reduced via this method, as with $3^{rd}$ kernel, the clock cycle is reduced from $61 \times 10^3$ seconds to $0.6 \times 10^3$ seconds, which means the result becomes $70.9\times$ faster. Meanwhile, according to the simulation test, this algorithm raises the ratio of actual DMA bandwidth to theoretical bandwidth up to above $92\%$, as shown in the $3^{rd}$ kernel, and even reaches the theoretical limit $95\%$ as a whole.

## 4.2 Mixed-precision Optimization

Introducing low-precision format data helps reduce memory use. Since the LDM in the slave kernel has limited storage, lowering the data precision could make better use of the given space and accelerate communication with the same actual DMA bandwidth. In terms of NEMO, due to its memory constraint, DMA data communication contributes to greater than $95\%$ of the total amount of running time on the many-core architecture, and as a result, we can enhance the overall performance by lowering the time cost of data communication. Focusing on tracer_fct in NEMO, we simulate a 4-kernel test with a 49x65x128 grid on average per kernel, and the results are shown in Figure 9, where the red and orange bars represent the average wall time of SP and SP+HP in one simulation. Since NEMO is sensitive to precision, we cannot use HP only. We can see that the SP+HP method shortens the total time by half compared to the original DP method. To further explore this topic, we simulate another test comparing the bandwidth usage the ratio in both SP and SP+HP precision, which is shown in Figure 10. The result shows that both methods surpass the $90\%$ ratio threshold, with the $4^{th}$ kernel performing best with a ratio of $95\%$.





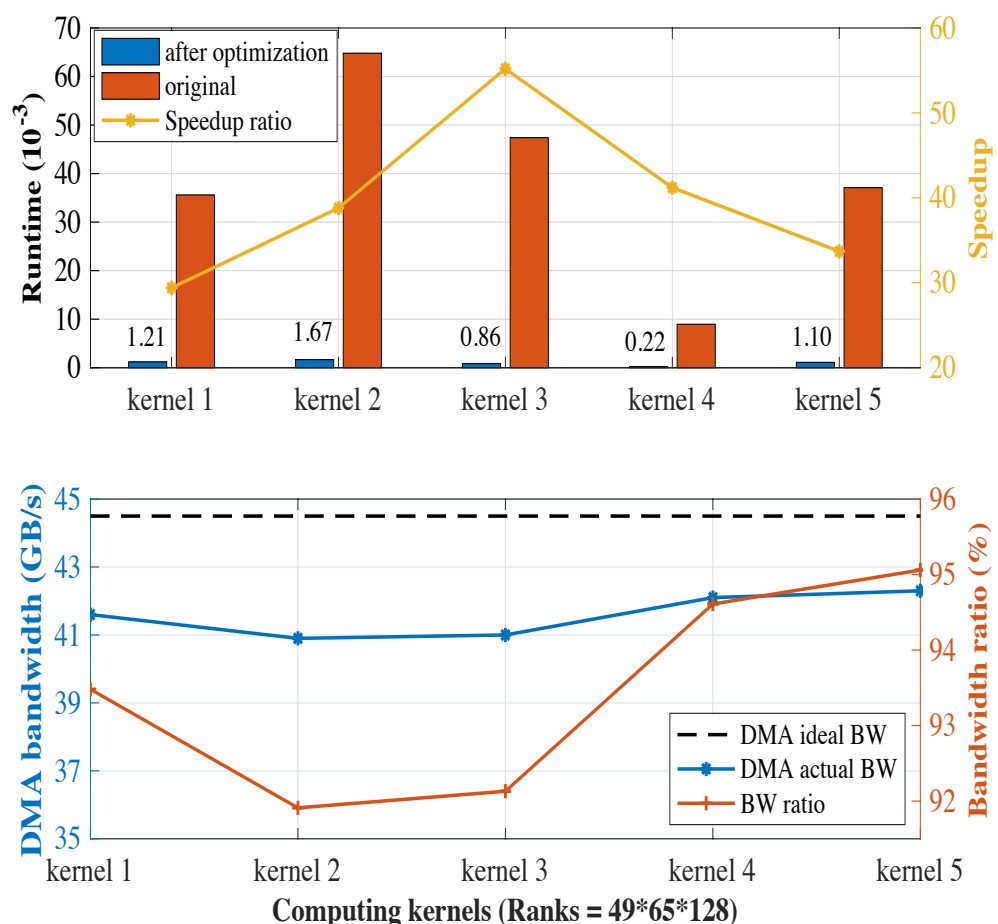

**Figure 8.** The performance of the 5-kernel simulation test and the 5-kernel are shown in the appendix

## 4.3  Strong Scaling

Figure 11 shows the results of strong scaling. We conduct experiments with resolutions of 2 km, 1 km and 500 m, increasing the number of cores from 2,129,920 to 27,988,480. The numbers of grids at resolutions of 2 km, 1 km and 500 m are 24,002×16,002×128, 43,502×29,002×128 and 82,502×55,002×128, respectively. We set 2,129,920 cores as the baseline of strong scaling, and the final parallel efficiencies are 74.18%, 83.40% and 99.29%, respectively.

When the number of cores is 2,129,920, the average computing task of each process (*i.e.*, CG) is approximately 325×432,
including a halo in the horizontal direction and 128 grids in the vertical direction. The mixed-precision optimization proposed in this paper considerably speeds up computations and reduces memory overhead. However, to guarantee the accuracy of the results, there are still some unavoidable double-precision floating-point arithmetic in the key steps in swNEMO_v4.0.



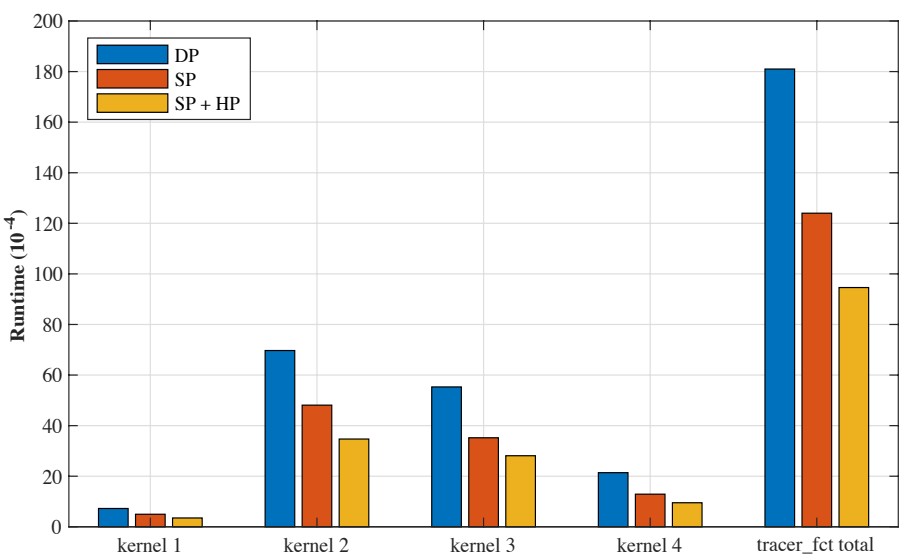

**Figure 9.** Time cost of the tracer_fct process

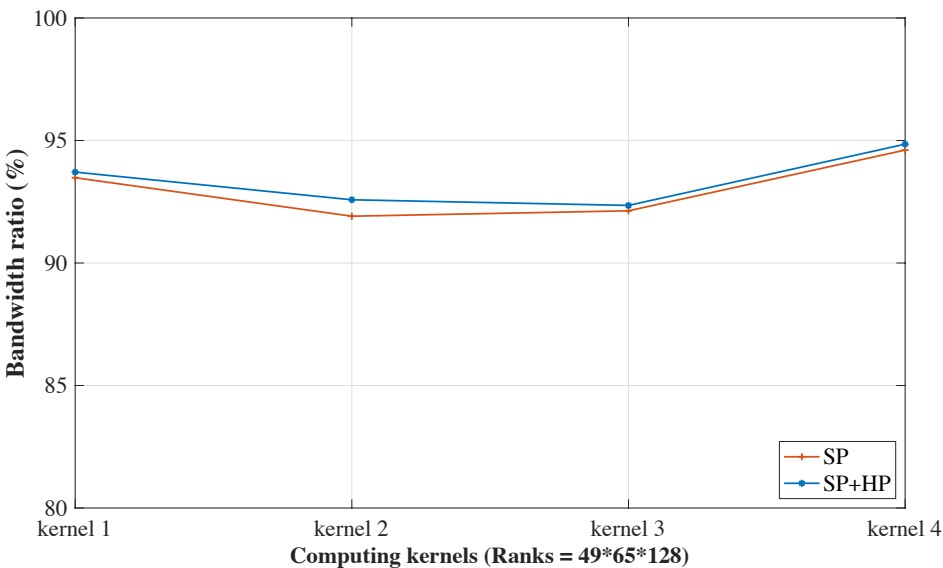

**Figure 10.** Bandwidth ratio of tracer_fct process

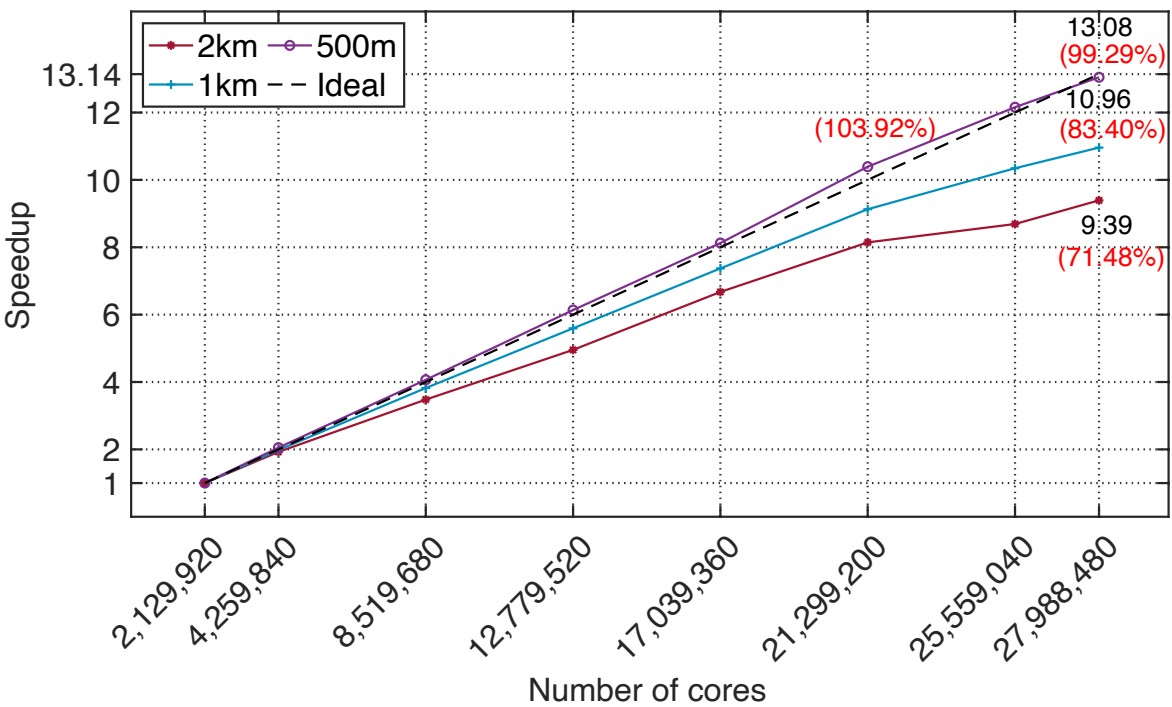

**Figure 11.** The strong scalability results, scaling from 2,129,920 to 27,988,480 cores with 2 km, 1 km and 500 m resolutions.

During computing, the size of a four-dimensional variable with double-precision has exceeded 1 GB. To maintain a reasonable efficiency of the memory system, we select 2,129,920 cores as the minimum available baseline.

In addition, there is a large number of stencil computations in swNEMO_v4.0. Assigning tasks reasonably can effectively reduce the amount of communication and balance in the computing tasks. The four-level parallelization approach fully coordinates the parallel loads of each level; thus, the super-linear speedup is finally achieved at a resolution of 500 m.

    For two-dimensional stencil computations, a grid with similar sizes of the $x$-axis and $y$-axis can effectively reduce the amount of communication and speed up the stencil computation. When the grid sizes of the $x$-axis and the $y$-axis are the

same, the amount of communication touches the bottom. According to the four-level parallel architecture, the process-level task assignment does not involve grid partitioning in the vertical direction. For the strong scalability test, with a resolution of 500 m, the size of the horizontal grid is $82,502 \times 55,002$, and the size of the vertical grid is 128. When using 21,299,200 cores, the process division in the horizontal direction is $640 \times 512$. At this time, each process (*i.e.* CG) computes $128 \times 108$ grids, and the grid sizes of the $x$-axis and the $y$-axis of each process are the closest in all options. Therefore, the best parallel efficiency,

104%, is achieved when the number of cores is 21,299,200.





In summary, the strong scalability of the three resolutions has always maintained high performance with ten million cores, and the speedup is still nearly linear at ultra-large scale. By using 27,988,480 cores, swNEMO_v4.0 achieves up to 99.29% parallel efficiency with a high resolution of 500 m.

### 4.4 Weak Scaling

The choice of resolution of the GYRE needs to follow strict disciplines. Therefore, to ensure that the workload in a single process is fixed, the number of grids of the *x*-axis and *y*-axis is proportional to the number of cores, and the automatic scheme of domain decomposition in swNEMO_v4.0 is replaced with a manual scheme at the same time, which is shown in Figure 12 and Table 3.

As we build roof-line models on NEMO, we find that when the horizontal grid surpasses $49 \times 65$ and the vertical grid sur-
passes 128, the computation efficiency of the floating point for a single core group performs the best. Therefore, our following experiments on weak scaling are conducted on a single core group with $49 \times 65 \times 128$ grid size each. Based on the above-mentioned principle, we choose resolutions of 9 km, 7 km, 4.5 km, 3.5 km, 2.5 km, 2.0 km, 1.2 km and 1.0 km. According to the expansion of the workload of each process (*i.e.*, CG), the number of cores increases from 299,520 to 27,988,480. With a resolution of 1.0 km, the total number of grids is 43,502×29,002×128. As shown in Figure 12, the performance is stable with
different resolutions, and the computation efficiency of the floating point is still 1.99‰the resolution of 1 km, which is very close to the baseline. The nearly linear trend indicates that the model has wonderful weak scalability.

### 4.5 Peak Performance

Figure 13 shows the peak performance of swNEMO_v4.0 with a resolution of 1 km. When the total grid size is 43,502×29,002×128, the number of cores increases from 2,129,920 to 27,988,480. It is obvious that the optimal performance is 1.97 PFlops by using
27,988,480 cores, and the performance of the optimized version is 10.25× faster than that of the original version.

In swNEMO_v4.0, we fully parallelize 70.58% of the hotspots according to the performance profiling tools. The remaining 29.42% of hotspots are mainly serial, which can hardly be parallelized. Therefore, the mix-precision optimization is used to optimize the serial part. According to Amdahl's law

$$\text{Speedup} = \frac{1}{(1-P)+P/N}$$

, the theoretical speedup is approximately 6.80×.

Combining the breakthroughs in an adaptive four-level parallelization design, CPE cluster optimization and mixed-precision optimization, the performance of our optimized version surpasses the theoretical values after further refactoring the code to reduce conditional judgement, instruction branching and the complexity of the code.

Additionally, as shown in Figure 14, 1.43 simulated years per day (SYPD) with a resolution of 1 km by using 27,988,480 cores is achieved, which is 7.53× compared with 0.19 SYPD of the original version, exceeding the ideal performance upper bound.



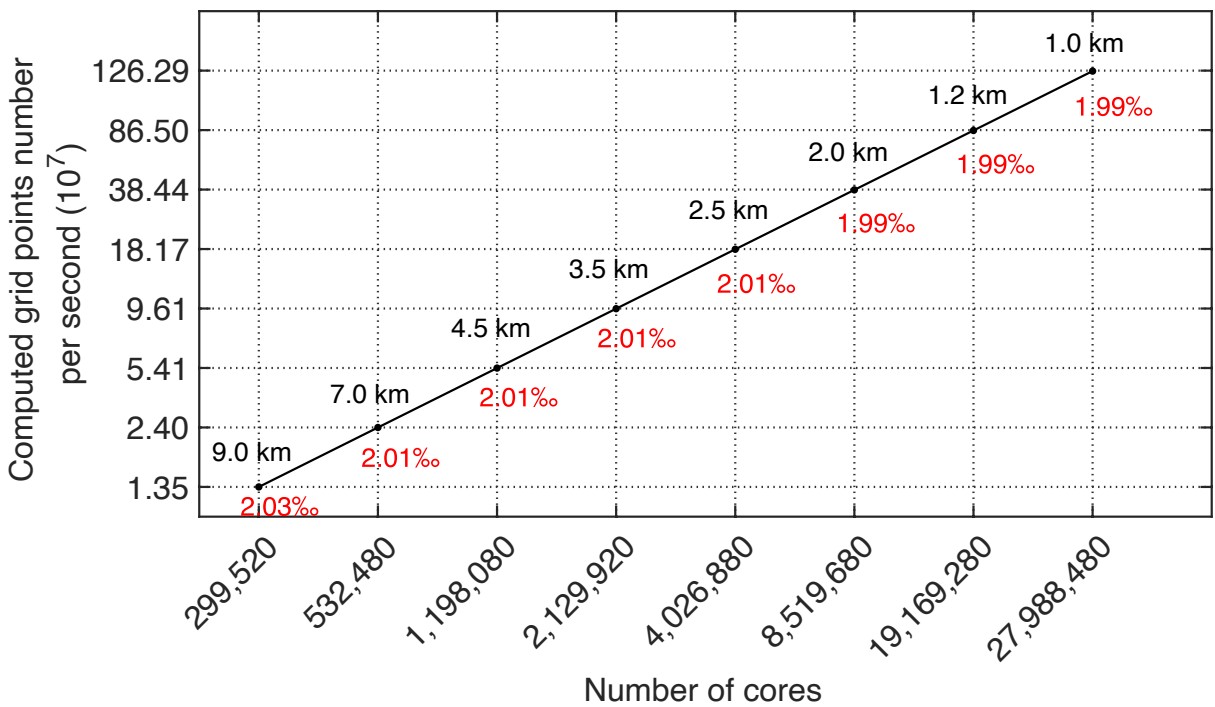

**Figure 12.** The weak scalability results, scaling from 299,520 to 27,988,480 cores with 8 different resolutions.

## 5   Conclusions and Discussions

This paper presents a successful solution for global ocean simulations with ultrahigh resolution using NEMO on a new gen-

eration Sunway supercomputer. Three breakthroughs, including an adaptive four-level parallelization design, many-core opti-
mization, and mixed-precision optimization, are designed and tested. The simulations achieve 71.48%, 83.40%, and 99.29%
parallel efficiency with resolutions of 2 km, 1 km, and 500 m by using 27,988,480 cores, respectively.

Current resolutions of ocean models in ocean forecasting systems and climate models cannot well resolve the meso and
submesoscale eddies in the real ocean, not to mention the other even smaller-scale processes such as internal waves. However,

these submesoscales and small-scale processes are quite important to navigation safety in themselves, and they also have
notable influences on the large-scale simulations of the ocean general circulations and global and regional climate through
the interacting with different scales. Improving resolution is one of the best ways to resolve these processes and improve
simulations, forecasts and predictions. The highest resolution used in this study is 500 m, and this resolution can well-resolve
the submesoscale eddies and partly resolve the internal waves, which are very important for the safety of offshore structures.

Therefore, the breakthroughs made in this study make the direct simulations of these important submesoscales and small-scale



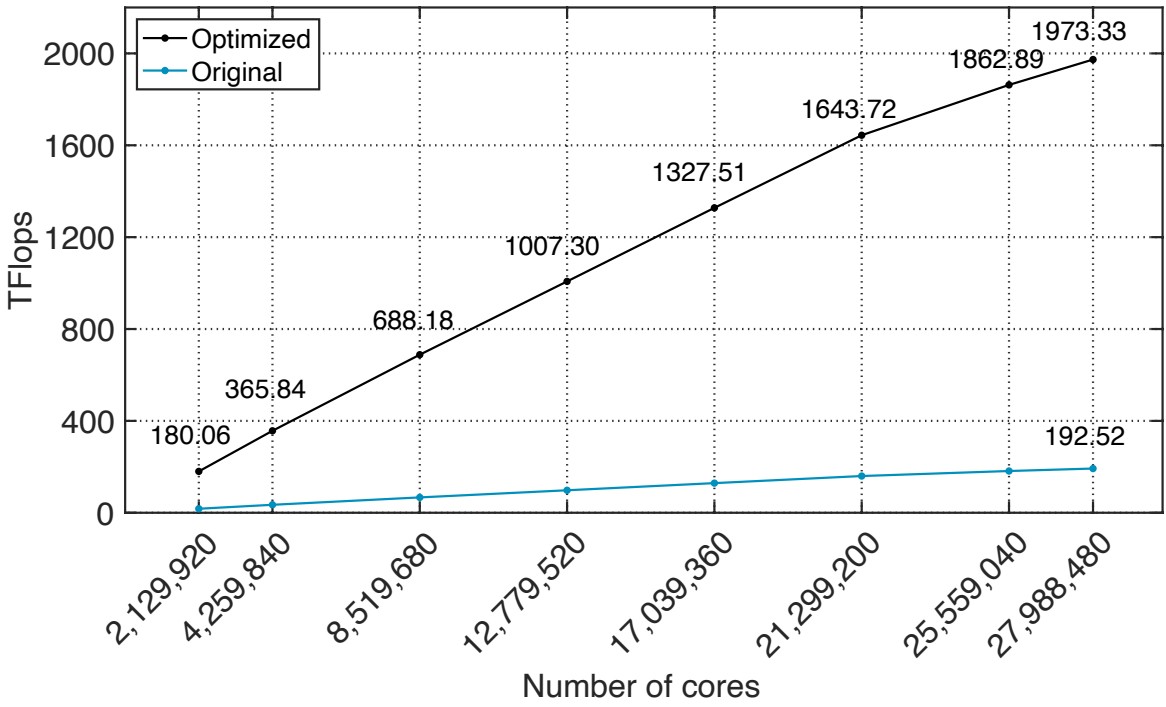

**Figure 13.** The peak performance results for simulations with 1 km resolution. Numbers along the graph line are the peak performances for simulations with different cores. The peak performance with 27,988,480 cores reaches 1.973 PFlops.

processes at the global scale possible. This will substantially improve the forecast and prediction accuracy of the ocean and climate, while strongly supporting the key outcome of a predicted ocean of the UN Ocean Decade.

This study is conducted in the new generation of Sunway supercomputers. The breakthroughs in this paper, such as the four-level parallelization design and the method for efficient data transportation inside CPEs, provide some novel ideas for
other applications in this series of Sunway supercomputers.

Moreover, our research indicates that mixed precision, especially half precision, is of great help for improving the efficiency of ocean simulations with high resolutions. However, existing CPU instructions and architecture have limited support for half-precision simulation, and CPE only supports simple computations, such as addition, subtraction, multiplication and division. We propose that the future design of the system should take half-precision computation into consideration.

High-resolution simulations of the ocean are crucially important for navigation safety, weather forecasting and global climate change prediction. We believe that these approaches are also suitable for other OGCMs and supercomputers. In this paper, three innovative algorithms proposed based on the new generation of Sunway supercomputers provide an important reference for ultrahigh resolution ocean circulation forecasting. However, only benchmarks are tested in the work, and real application data



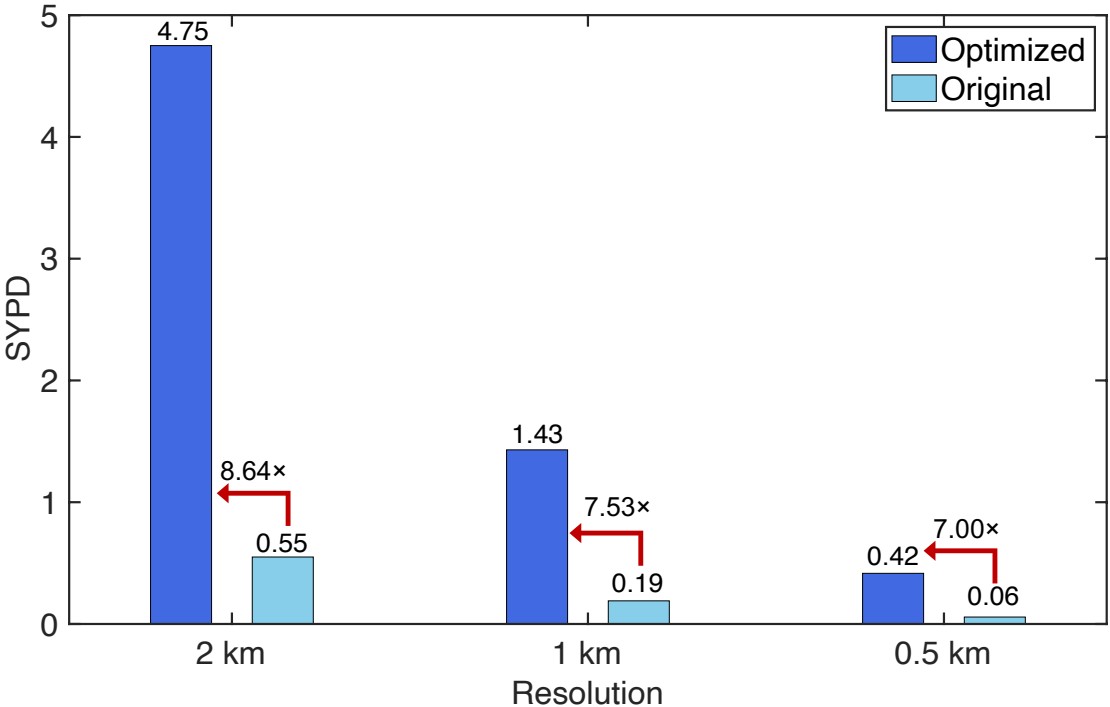

**Figure 14.** The computing throughput results by using 27,988,480 cores. SYPD increases from 0.19 to 1.43 with the 1 km resolution. The optimized performance is $7.53\times$ that of the original version.

has not been used. In the future, we will build an ultrahigh-resolution ocean circulation forecasting model under real scenarios

and conduct in-depth research to provide efficient solutions to accurately predict ocean circulation and climate change.

*Code availability.* The swNEMO_v4.0 source codes with user manual are available at https://doi.org/10.5281/zenodo.5976033 (Ye et al., 2022).

## Appendix A: Supplementary code in Fortran format

The following five chunks of code correspond to the five kernels stated in our previous experimental section. The codes listed

below are exactly from NEMO for experimental validation.

**Kernel 1**

$$\textbf{DO } jk = 2, jpkm1$$
$$\quad \textbf{DO } jj = 2, jpjm1$$
$$\quad\quad \textbf{DO } ji = fs\_2, fs\_jpim1$$





$$zmsku = wmask(ji,jj,jk)/MAX(umask(ji,jj,jk-1)+umask(ji-1,jj,jk)$$
$$\& + umask(ji-1,jj,jk-1)+umask(ji,jj,jk),1.0)$$
$$zmskv = wmask(ji,jj,jk)/MAX(vmask(ji,jj,jk-1)+vmask(ji,jj-1,jk)$$
$$\& + vmask(ji,jj-1,jk-1)+vmask(ji,jj,jk),1.0)$$
$$zahu\_w = (pahu(ji,jj,jk-1)+pahu(ji-1,jj,jk)$$
$$\& + pahu(ji-1,jj,jk-1)+pahu(ji,jj,jk))*zmsku$$
$$zahv\_w = (pahv(ji,jj,jk-1)+pahv(ji,jj-1,jk)$$
$$\& + pahv(ji,jj-1,jk-1)+pahv(ji,jj,jk))*zmskv$$
$$ah\_wslp2(ji,jj,jk) = zahu\_w*wslpi(ji,jj,jk)*wslpi(ji,jj,jk)$$
$$\& + zahv\_w*wslpj(ji,jj,jk)*wslpj(ji,jj,jk)$$
**END DO**

**END DO**

**END DO**

---

**Kernel 2**

**DO** $jj = 2, jpjm1$

**DO** $ji = fs_2, fs_jpim1$
$$pta(ji,jj,1) = e3t_b(ji,jj,1)*ptb(ji,jj,1)+p2dt*e3t_n(ji,jj,1)*pta(ji,jj,1)$$
**END DO**

**END DO**

**DO** $jk = 2, jpkm1$

**DO** $jj = 2, jpjm1$

**DO** $ji = fs_2, fs_jpim1$
$$zrhs = e3t_b(ji,jj,jk)*ptb(ji,jj,jk,jn)+p2dt*e3t_n(ji,jj,jk)*pta(ji,jj,jk)$$
$$pta(ji,jj,jk) = zrhs - zwi(ji,jj,jk)/zwt(ji,jj,jk-1)*pta(ji,jj,jk-1)$$
**END DO**

**END DO**

**END DO**

---

**Kernel 3**

**DO** $jj = 2, jpjm1$

**DO** $ji = fs_2, fs_jpim1$

$$zwt(ji,jj,1) = zwd(ji,jj,1)$$
**END DO**

**END DO**

**DO** $jk = 2, jpkm1$

**DO** $jj = 2, jpjm1$





**DO** $ji = fs_2, fs_j pim1$

$$zwt(ji, jj, jk) = zwd(ji, jj, jk) - zwi(ji, jj, jk) * zws(ji, jj, jk - 1)/zwt(ji, jj, jk - 1)$$

     **END DO**

     **END DO**

     **END DO**

**DO** $jk = 1, jpkm1$

     **DO** $jj = 2, jpjm1$

       **DO** $ji = fs_2, fs_j pim1$

       $zwi(ji, jj, jk) = -p2dt * zwt(ji, jj, jk)/e3w_n(ji, jj, jk)$

       $zws(ji, jj, jk) = -p2dt * zwt(ji, jj, jk + 1)/e3w_n(ji, jj, jk + 1)$

405       $zwd(ji, jj, jk) = e3t_a(ji, jj, jk) - zwi(ji, jj, jk) - zws(ji, jj, jk)$

       **END DO**

     **END DO**

   **END DO**

---

**Kernel 4**

---

**DO** $jk = 1, jpkm1$

     **DO** $jj = 1, jpjm1$

       **DO** $ji = 1, fs_j pim1$

       $zdit(ji, jj, jk) = (ptb(ji + 1, jj, jk) - ptb(ji, jj, jk)) * umask(ji, jj, jk)$

       $zdjt(ji, jj, jk) = (ptb(ji, jj + 1, jk) - ptb(ji, jj, jk)) * vmask(ji, jj, jk)$

415       **END DO**

     **END DO**

   **END DO**

   **DO** $jk = 1, jpkm1$

   $zdk1t(:, :) = (ptb(:, :, jk) - ptb(:, :, jk + 1)) * wmask(:, :, jk + 1)$

**IF** $jk == 1$ **THEN** $zdkt(:, :) = zdk1t(:, :)$

   **ELSE**

     **DO** $jj = 1, jpj$

       **DO** $ji = 1, jpi$

       $zdkt(ji, jj) = (ptb(ji, jj, jk - 1) - ptb(ji, jj, jk)) * wmask(ji, jj, jk)$

425       **END DO**

     **END DO**

   **END IF**

   **DO** $jj = 1, jpjm1$

     **DO** $ji = 1, fs_j pim1$




$zabe1 = pahu(ji,jj,jk) * e2_e1u(ji,jj) * e3u_n(ji,jj,jk)$

$zabe2 = pahv(ji,jj,jk) * e1_e2v(ji,jj) * e3v_n(ji,jj,jk)$

$zmsku = 1./MAX(wmask(ji+1,jj,jk) + wmask(ji,jj,jk+1)$

$\& + wmask(ji+1,jj,jk+1) + wmask(ji,jj,jk),1.)$

$zmskv = 1./MAX(wmask(ji,jj+1,jk) + wmask(ji,jj,jk+1)$


$\& + wmask(ji,jj+1,jk+1) + wmask(ji,jj,jk),1.)$

$zcof1 = -pahu(ji,jj,jk) * e2u(ji,jj) * uslp(ji,jj,jk) * zmsku$

$zcof2 = -pahv(ji,jj,jk) * e1v(ji,jj) * vslp(ji,jj,jk) * zmskv$

$zftu(ji,jj,jk) = (zabe1 * zdit(ji,jj,jk)$

$\& + zcof1 * (zdkt(ji+1,jj) + zdk1t(ji,jj)$


$\& + zdk1t(ji+1,jj) + zdkt(ji,jj))) * umask(ji,jj,jk)$

$zftv(ji,jj,jk) = (zabe2 * zdjt(ji,jj,jk)$

$\& + zcof2 * (zdkt(ji,jj+1) + zdk1t(ji,jj)$

$\& + zdk1t(ji,jj+1) + zdkt(ji,jj))) * vmask(ji,jj,jk)$

**END DO**

**END DO**

**DO** $jj = 2, jpjm1$

   **DO** $ji = fs_2, fs_jpim1$

   $pta(ji,jj,jk,jn) = pta(ji,jj,jk,jn) + zsign * (zftu(ji,jj,jk) - zftu(ji-1,jj,jk)$

   $\& + zftv(ji,jj,jk) - zftv(ji,jj-1,jk))$

$\& * r1_e1e2t(ji,jj)/e3t_n(ji,jj,jk)$

   **END DO**

   **END DO**

**END DO**

---

**Kernel 5**

**DO** $jk = 2, jpkm1$

   **DO** $jj = 2, jpjm1$

      **DO** $ji = fs_2, fs_jpim1$

      $zmsku = wmask(ji,jj,jk)/MAX(umask(ji,jj,jk-1) + umask(ji-1,jj,jk)$

      $\& + umask(ji-1,jj,jk-1) + umask(ji,jj,jk),1.0)$

$zmskv = wmask(ji,jj,jk)/MAX(vmask(ji,jj,jk-1) + vmask(ji,jj-1,jk)$

      $\& + vmask(ji,jj-1,jk-1) + vmask(ji,jj,jk),1.0)$

      $zahu_w = (pahu(ji,jj,jk-1) + pahu(ji-1,jj,jk)$

      $\& + pahu(ji-1,jj,jk-1) + pahu(ji,jj,jk)) * zmsku$

      $zahv_w = (pahv(ji,jj,jk-1) + pahv(ji,jj-1,jk)$



$\quad \& + pahv(ji,jj-1,jk-1) + pahv(ji,jj,jk)) * zmskv$

$\quad zcoef3 = -zahu_w * e2t(ji,jj) * zmsku * wslpi(ji,jj,jk)$

$\quad zcoef4 = -zahv_w * e1t(ji,jj) * zmskv * wslpj(ji,jj,jk)$

$\quad ztfw(ji,jj,jk) = zcoef3 * (zdit(ji,jj,jk-1) + zdit(ji-1,jj,jk)$

$\quad \& + zdit(ji-1,jj,jk-1) + zdit(ji,jj,jk))$

$\quad \& + zcoef4 * (zdjt(ji,jj,jk-1) + zdjt(ji,jj-1,jk)$

$\quad \& + zdjt(ji,jj-1,jk-1) + zdjt(ji,jj,jk))$

$\quad ztfw(ji,jj,jk) = ztfw(ji,jj,jk) + e1e2t(ji,jj)/e3w_n(ji,jj,jk) * wmask(ji,jj,jk)$

$\quad \& * (ah_w slp2(ji,jj,jk) - akz(ji,jj,jk))$

$\quad \& * (ptb(ji,jj,jk-1) - ptb(ji,jj,jk))$

$\quad$ **END DO**

$\quad$ **END DO**

$\quad$ **END DO**

---

*Author contributions.* Y. Ye, Z. Song, and S. Zhou: Methodology, coding, numerical experiment, analysis, and writing–original draft prepa-
ration; Y. Liu, Q. Shu, and B. Wang: Algorithm description, and validation; Y. Liu and W. Liu: Algorithm and numerical experiments' sug-
gestion; F. Qiao, and L. Wang: Conceptualization, supervision, funding acquisition, and writing–review and editing; All authors discussed,
read, edited, and approved the article. All authors have read and agreed to the published version of the manuscript.

*Competing interests.* The authors have declared that neither they nor their coauthors have any competing interests.

*Acknowledgements.* This work is supported by the National Natural Science Foundation of China (U1806205, 42022042, and 41821004)
and CAS Interdisciplinary Innovation Team (JCTD-2020-12).





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
