# Peer review of "swNEMO\_v4.0: an ocean model NEMO4 for the next generation Sunway supercomputer"

_Geoscientific Model Development, 2022_

## Author Response (AR1)

Dear Editors:

Thank you for your work. We also appreciate the suggestions and comments from the two reviewers on our manuscript (Paper #  GMD-2022-33) entitled: "swNEMO_v4.0: an ocean model NEMO for the next generation Sunway supercomputer". In the revision, we have tried our best to consider and incorporate all suggestions and comments provided by the three reviewers. The point-by-point responses to the two reviewers' suggestions and comments are attached. As we run the perturbation experiments to validate the mixed-precision optimized results following the reviewer's suggestion, we did not finish the work until now. Thank you for your patience.

Sincerely yours,

Fangli Qiao, on behalf of all authors

**§1 Response to Reviewer #1**

(Note: Referee comments in black, ***reply in bold italics***)

This paper develops a unique ultrahigh scalability parallelized NEMO ocean model on the Sunway supercomputer architecture. A new many-core optimization using remote memory access (RMA) blocking and dynamic cache scheduling can effectively accelerate the performance more 90% of ideal bandwidth. The strategical optimization based on mixed precision improves the parallel performance to achieve more than 99% with appropriate 28 million cores. This represents significant progress in the ocean modelling parallelization. The impact will be tremendous. However, there are two major issues to be addressed.

***Reply: We would like to express our sincere thanks for your valuable comments. The revised manuscript has been refined according to your suggestions. These comments and suggestions greatly help us in improving the quality of this manuscript.***

Major issues:

1. A very important aspect of improving the parallel performance is to ensure the reproducibility. This study provide a significant speed up combining hardware and software optimization. However, using the mixed precision can change the solution if different cores are used? Can the mixed precision affect the reproducibility and consistency? The authors should address and discuss this issue to ensure the robustness of the proposed model.

***Reply: Thank you for your valuable suggestions. Firstly, this work proposed the optimization algorithm for NEMO, based on the new generation Sunway platform. We tested several times and the results can be reproduced, even with different processes. Although this work focused on the NEMO and Sunway platform, the algorithms, such as longitude-latitude-depth, double-single-half mixed precision, are general methods and can be used for other OGCMs or platforms. Certainly, as we used several features of new generation Sunway platform (e.g., the RMA commination between CPEs, half-precision in CPEs) enabling the efficiency, the speed up may become lower in other platforms. Secondly, we ran the error growth tests based on the method proposed by Baker et al. 2016 after the mixed-precision optimization. Yes, the mixed-precision affects the results. And the***

*effects are similar with these of the perturbation coefficient as O(10^-10) (Fig r1), which*
*can be accepted we think.*

*Following your suggestion, we added the discussion on the tests in section 3.3.*

[Figure]

**Fig r1. Z-score biases of the sea surface temperature in ensemble run (grey lines) with**
**perturbation coefficient as O(10^-10)  and the mixed-precision experiment (red line)**

2. The other issue is related to the commonly used ensemble simulation while different
precision is used. Baker et al. (2016) evaluated the consistency and proposed the perturbation
requires half precision (see the large variation of SST simulation in Fig. 3 of Baker et al.,
2016). The mixed-precision OGCM can causes the Bit-to-bit inconsistency within ocean
model. Is that correct? How can this MP approach compare with the reduced convergence
accuracy in the solver, which can also speed up the simulation?

*Reply: Thank you for the constructive suggestion. Yes, the mixed-precision can cause the*
*bit-to-bit inconsistency within ocean model. In order to verify the correctness of the mixed-*
*precision results, we provide the perturbation simulation experiment for the temperature*
*results following Baker et al. (2016), in which random disturbance conditions for the initial*

*temperature are configured within NEMO program. At the same time, we also selected 101*

*sets of data as experimental simulation results to verify the correctness of the results of*

*NEMO_v4.0, of which 100 sets are temperature results with perturbation conditions, and*

*the remaining one is the mixed-precision experiment of swNEMO_v4.0.*

*According to the Baker et al. (2016), we also selected $O(10^{-14})$ as the perturbation*

*coefficient, but found that the Z-score obtained by mixed-precision simulation was*

*completely out of the shadow area formed by 100 simulations with double-precision (Fig*

*r2). Therefore, we increased the perturbation coefficient and he mixed-precision Z-score*

*gradually approached the shadow area. We found that the disturbance had a greater*

*influence in the first few years, then it gradually decreased and tended to be stable after*

*several years(Fig r2, r3). As shown Fig r3, when the perturbation coefficient is $O(10^{-11})$,*

*Z-score of mixed-precision simulation fall partially in the region formed by double-*

*precision simulations. When the perturbation coefficient equals to $O(10^{-10})$, the Z-score*

*completely falls in the shadow area (Fig r1), which indicates the effects of mixed-precision*

*are similar with these of the perturbation coefficient as $O(10^{-10})$.*

*Following your suggestions, we added the discussion on the tests in section 3.3.*

[Figure]

*Fig r2. Z-score biases of the sea surface temperature in ensemble run (grey lines) with*

*perturbation coefficient as $O(10^{-14})$ and the mixed-precision experiment (red line)*

[Figure]

*Fig r3. Z-score biases of the sea surface temperature in ensemble run (grey lines) with perturbation coefficient as O(10^-11) and the mixed-precision experiment (red line)*

Minor issues:

1. Line 7, Abstract: DMA is not defined. what do you mean by DMA? Do you refer to remote memory access (RMA) or something else (Direct memory access)?

*Reply: Thank you. DMA means Direct memory access. The definition is added in the revised abstract.*

2. Line 21, change "the one of most important directions of OGM development" to " one of the most important directions for the OGCM development".

*Reply: Thank you. It was refined.*

3. Line 23, change "horizontal resolution doubled" to "doubled horizontal resolution".

*Reply: Thank you. It was refined.*

4. Line 31, what do you mean by 6.8x? Do you mean by a factor of 6.8? If so, I suggest to change this rather than symbol x. This can be seen elsewhere.

*Reply: Thank you. 6.8x means 6.8 times. It was refined in the revision.*

5. Line 31, "achieved the performance of 408 Intel Westmere cores on four K20 GPUs". What do you mean by this? What performance is achieved? Equivalent performance of 408 Intel Westmere cores using 4 K20 GPUs? However, how many gpu cores for the K20 GPUs? The cores of Intel processors are not equivalent to the cores of GPU processors, right?

*Reply: Thank you. Yes, for the calculation example of POM designed on GPU, the computing performance by using 4 K20 GPUs is equivalent to the performance by using 408 Intel Xeon x5670 CPUs.*

*It was refined as "achieved the equivalent performance of 408 Intel Westmere cores by using four K20 GPUs" for clarity in the revision.*

6. Line 27-43, Table 1 and the review of performance improvement are impressive. However, are they all for the improvement of ocean models? FUNWAVE seems to be a wave model? What about MUSNUM? I suggest to separate wave model to a different category since the architecture of a wave model is totally different from the ocean dynamical model. Also, what's the difference between POP2 and CESM-HR? It seems they are both 3600x2400 resolution, right? While the performances are similar but the maximum scales quite different (~4 times). I suggest to tabulate the representative ocean model performance development here (exclude other types of models) and discuss the most significant development.

*Repy: Thank you for the suggestions. The Hydrostatic LAM is the atmosphere model. FUNWAVE and MASNUM are both the ocean surface wave model. CESM-HR is the high-resolution version of CESM, which including the POP ocean model. We tried to give progress on the numerical earth model, including the atmosphere model and ocean model at first. However, we total agreed with you that it can confuse the reader indeed. Therefore, wea remove the Hydrostatic LAM, FUNWAVE, MASNUM, and CESM-HR and only leave the progress on OGCMs. And we also changed the title of Table 1 to "Research on the OGCMs based on heterogeneous architectures" in the revision.*

7. Line 44-54, the discussion here also mixes the parallelization of atmosphere model, ocean hydrodynamic model and ocean wave models. Particularly, the required global barriers are also different. This can significantly impact the model overall performance. Don't mix the

ocean hydrodynamical model with other types of model in the comparison because the solvers are totally different. Also, this paragraph mixes the different limitations of different models to improve their performance without specific focus. I suggest to reorganize this discussion to be more focused and related to the improvement relevant to this study.

*Reply: Thank you the valuable suggestions. We total agree with you. It may confuse the reader. Therefore, we only leave the OGCMs and re-organized this paragraph in the revision.*

8. Line 46, change "only improved" to "is only improved".

*Reply: Thank you. It was refined.*

9. Line 56, change "Exa-scale to "Exascale".

*Reply: Thank you. It was refined.*

10. Line 78, is "GYRE-PISCES" abbreviation? If it is not a well-known typical benchmark test name, I suggest to described this briefly here or used a whole name.

*Reply: Thank you. GYRE-PISCES is the benchmark abbreviation of the Gyre Pelagic Interactions Scheme for Carbon and Ecosystem Studies. The detail of GYRE-PISCES is described in section 4, therefore we added the whole name of GYRE-PISCES here in the revision.*

11. Section 2 describes the architecture of Sunway TaihuLight. The detailed information has been provided extensively. I suggested to remove the technical details but comment and address on the specific features facilitating the performance enhancement used in this paper here.

*Reply: Thank you. In the revision, we removed "MPE is based on SW64 instruction set,with a 32KB L1 instruction cache, a 32KB L1 data cache and 512KB L2 cache."*

12. Section 2 also describes NEMO model. What's the difference between NEMO and NEMO4 you raised at line 81? I suggest to move NEMO description into section 3 in associated the porting of NEMO.

*Reply: Thank you. In the revision, we moved NEMO description into section 3 in associated the porting of NEMO.*

13. Line 120, how "adaptive" works in this four level parallelization? Two levels are using domain decomposition. One level is MPE-CPE asynchronous parallel. Is this performed at compiler level (processor specific) or user specific level? One level is the vector reconstruction. This should be done within the compiler level. Can the author comment which level contributes mostly to help the performance in the current implementation?

*Reply: Thank you. The main contribution of this optimization scheme comes from the division of tasks to the CPEs, which enable the model using the slave processer. In terms of adaptability, SW26010pro has heterogeneous architecture, which is different from the traditional homogeneous CPU architecture. MPE-CPE parallelization cannot be automatically implemented by the compiler. Code-level optimization is required to achieve specific data division and efficient transmission. Therefore, we re-consider the task division and parallelism according to the hardware characteristics of Sunway to fully using the computing capacity. Vectorization parallelism can usually be completed automatically by the compiler, but for complicated code logic, the compiler cannot be well optimized, so we manually implemented vectorization code to achieve better performance.*

*In the revision, we added the explanation on the different parallelization schemes to clarify clearly in section 3.*

**14. Line 130,** a reference is helpful for this MPE-CPE asynchronous parallelization. As described in line 131, IO can be independently separated for sure. However, how boundary data exchange can be parallelized aside from the computation? Normally, the ocean model kernel requires some global communication to solve the pressure equation (normally at least 3, can reduced to 1 in some parallelization). How can the data exchange be performed using MPE-CPE asynchronous parallelization. Some information will be helpful for the readers.

*Reply: Thank you for your constructive suggestions. We added the references in the revision. Gu et al. (2022, doi:10.1016/j.scib.2022.03.009) proposed multi-level optimization method to make use of the heterogeneous architecture. In the first level of the method, they designed pre-communication, communication, and post-communication in the code based on MPE-CPE asynchronous parallelism architecture. As the default barotropic solver in NEMO4 is explicit method with small time step, which is more accurate and more suitable for high resolution without filter, instead of implicit or split-explicit methods (e.g., PCG), there is no global communication. Therefore, it is only necessary to update the information of the halo region between different processes. Moreover, in explicit method, most boundary information exchanges are independent of the partition data in the process and can be used for asynchronous parallelization.*

*Following your suggestions, the description of boundary data exchange features is added in section 3.1.2 in the revision.*

15. 1.3, for latitude-depth decomposition, since this depth is not parallel friendly dimension. The parallelization requires level dependence. That means if the depth dimension is changed, the user needs to adjust something for LDA. Is that correct?

*Reply: Thank you. As we design the dynamical block size methodology based on the level number and solve the data dependency problem by exchanging boundary data via the RMA (detail please find in Section 3.2.1), there is no need to adjust codes or parameters for LDA if the depth dimension is changed.*

16. Line 171-line 174, What is alpha_1, beta_2 and beta_1, beta_2 within the equations. The notations are not standard mathematically. Is f a function? or a value represented by the 2nd line? These equations should be labeled numbers. What is x? is x an array? Please rewrite the formula in a more mathematical way?

*Reply: Thank you for your suggestions. We forgot it. In fact, alpha_1, beta_1, and beta_2 are only the coefficient. And f is the mathematical formula for the loop segment, x means the x-axis (longitude axis) in the coordinate system. We revised the formula in the revision.*

17. 2 discusses the optimization used here. It seems 3.2.1 is used as level 3 described in 3.1. Is that correct? Or combining the 4 level parallelization? Is 3.2.2 used in the MPE-CPE asynchronous parallel or something different? If so, I suggest to reorganize this discussion and make this clear. Section 3.3 discusses the mixed precision optimization, which I believe is different from the four level parallelization. Also, line 108-113, describes three major contributions while the 2nd one is used within the 1st four-level parallelization, right?

*Reply: Thank you. The section 3.1 introduces mainly the design of the overall parallel framework of NEMO, which emphasizes the connection among different levels in the parallel framework and the systematic framework constructed for the unique on-chip heterogeneity of Sunway. The section 3.2.1 described the paralleling-optimization scheme of local computing hot-spots based on Sunway. In other words, section 3.1 presents a general view of optimization, while section 3.2.1 gives the optimization in detail for local computing hot-spots.*

*In the revision, we added the brief introduction to these optimization scheme before section 3.1.*

18. Line 216, the maximum biases reach 0.05%. Are these biases the deviation between DP and HP? However, considering the chaotic behavior with time, can this bias propagate? Can the biases become larger with time? If this is the case, can the model result get bit-to-bit consistency which is a very important feature for ocean model within an earth system model? For the pressure solver within the ocean dynamical kernel, do you still use DP? If you still use DP, the convergence will still take time. Can you compare this optimization with another easier way by reducing the pressure solver criteria to a lower level (change from $10^{-13}$ to $10^{-7}$)? Changing the pressure solver criteria to a lower level can significantly reduce the computational time. Why not just use this simple approach since you already reduce the precision? Do I miss something? Normally for the ocean model, the most intensive computational cost is the pressure solver rather than the tracer equation, right? Why not use this approach while still preserving the precision?

Reply: ***Thank you for the constructive suggestion.***

*On the one hand, the mixed-precision can cause the bit-to-bit inconsistency within ocean model. In order to get the tolerance of the mixed-precision results, we provide the perturbation simulation experiment for the temperature results following the Baker et al. (2016). According to the Baker et al. (2016), we also selected O(10^-14) as the perturbation coefficient, but found that the Z-score obtained by mixed-precision simulation was completely out of the shadow area formed by 100 simulations with double-precision. Therefore, we adjusted the perturbation coefficient. The mixed-precision Z-score gradually approached the shadow area when increasing the perturbation coefficient. In the process, we found that the disturbance had a greater influence in the first few years, then it gradually decreased and tended to be stable after several years(Fig r2, r3). As shown Fig r3, when the perturbation coefficient is O(10^-11), Z-score of mixed-precision simulation falls partially in the region formed by double-precision simulations. The mixed-precision Z-score completely fell in the shadow area when the perturbation coefficient equals to O(10-^10) (Fig r1), which indicates the tolerance of mixed-precision is around O(10^-10).*

*On the other hand, as the default barotropic solver in NEMO4 is explicit method with small time step, which is more accurate and more suitable for finer resolution without filter, instead of implicit or split-explicit methods (e.g., PCG). According to analysis of hotspot testing, we found that the most intensive computational cost is the tracer equation.*

*The above description was added in the Conclusion and Discussion section in the revision.*

19. Line 223, change "periodical" to "periodic". What do you mean by "North Pole folding"? Do you mean "Displaced North Pole"?

*Reply: Thank you. It's tripolar grid, not the dual-polar grid with Displaced North Pole.*

20. Line 230, change "is equal to" to "equals to".

*Reply: Thank you. It was refined.*

21. Line 229-234, this paragraph is confusing. It describes "three experiments with 2 km, 1 km, and 500m". However, each experiment uses 8 different parallel scales (Table 3),

resolution ranging from 9km to 1km. Do you use 2km, 1km and 500m or 9km to 1km? I suggest to clarify these numerical experiments. What's your definition of weak scaling and strong scaling.

*Reply: Sorry for confusing you. Besides the strong scalability experiments with 2 km, 1 km, and 500 m, we also carried out the weak scalability experiments. The strong scaling is tested by running the model with different numbers of threads, while keeping the same resolution. The weak scaling is tested by running the model with a fixed grids-per-thread ratio. In other words, the weak scaling means keep similar grids per thread. Therefore, we should fine the resolution with increasing used threads (Table 3).*

22. Line 242, what is "CPEs parallel method"? Is this your control experiment? This has nothing to do with the MPE-CPEs parallelization, right? However, does CPEs parallelization still use four-level parallelization? Can you isolate the individual performance enhance resulting from the approaches discussed in section 3?

*Reply: Thank you for your suggestions. CPEs parallel method is the third level of the four-level parallelism method, which is the parallelization of 64 CPEs, while MPE-CPES is the second level parallelism, which regards the CPEs as a component, and it is described the parallelism between the component and the main processing element.*

*In the revision, we added the explanation on the different parallelization schemes to clarify clearly.*

23. Line 248-253, do you include the performance increase due to the mixed-precision approach here or just the DMA and FLOPS for the DP? The timing may be different.

*Reply: Thank you. The performance results are from the model based on all the optimizations, including the four level and mixed-precision approaches.*

*In the revision, we added the explanations to clarify clearly.*

24. Line 256, can you describe these five kernels briefly? What's the major differences?

*Reply: Thank you. In fact, the five kernels are the most time-consuming chunks, which are only a fraction of the physical process. However, they are all specific implementations of*

*Stencil computing. To avoid confusing reader, we added the explanations for five kernels in the revision.*

25. 8, do you use the real time? Or measure the clock? These are built-in hardware, is it right? Therefore, these values only refer to the access time, right?

*Reply: Sorry for confusing you. It is the clock time measured by built-in hardware. In the revision, we revise the y-axis of Fig. 8 from Runtime to Clock time. We made a small typo in the last paragraph of section 4.1. The sentence "the clock cycle is reduced from 61 x 10^3 seconds to 0.6 x 10^3 seconds" has been changed to "the clock cycle is reduced from 61 x 10^-3 seconds to 0.6 x 10^-3 seconds".*

26. Section 4.2, is the implementation only performed for the tracer equations? Fig. 9 shows only the tracer integration which is only a very small portion of the overall run time. Can the author show the dynamical solver part which requires the most intensive computation (particularly the barotropic solver) instead of this tracer solver?

*Reply: Thank you. We run the performance-testing tool and found that one of the most time-consuming subroutines is tracer integration, so we focus on it for optimization. We think the barotropic solver being changed in NEMO4 leads to different bottleneck. The default barotropic solver in NEMO4 is explicit method with small time step, which is more accurate and more suitable for finer resolution without filter, instead of implicit or split-explicit methods (e.g., PCG, needing a lot of global communication). As there is no global communication and only necessary to update the information of the halo region between different processes, the barotropic solver is not the bottleneck anymore.*

27. Since this is GMD rather than computational journal, can the authors show the final results? It will be useful to examine if the GYRE-PISCES configuration reaches the expected solution as others. A figure with velocity and temperature fields will be enough, particularly what specific features can be found at 500m resolution. The potential impact of mixed precision optimization can also be discussed.

*Reply: Thank you for the constructive suggestions. Yes, it will be better if we can show the results with 2 km, 1km or 500 m resolutions. However, the output is too huge! We tried to store the results with 1 km resolution, but the data volume is more than 65 TB per output. Therefore, we did not store the output. And it's also the reason we did the validation for mix-precision by using the low resolution (0.5-degree). The experiment shows that the mixed-precision affects the results, but when the perturbation coefficient is around $O(10^{-10})$, effects are very small and can be accepted.*

28. Line 286, the description is very superficial, any supporting evidence?

*Reply: Thank you. It is a general description and summary for the following paragraph. To avoid confusion, we remove this paragraph in the revision.*

**§2 Response to Reviewer #2**

(Note: Referee comments in black, ***reply in bold italics***)

This work developed an ocean model called swNEMO_v4.0 based on a new-generation Sunway supercomputer and obtained significant modeling performance by sophisticated tuning methods that fully exploited the computing recourses of the new machine. Optimizing methods proposed are based on the architectural features, and thus achieves promising modeling performance. Thread-level communication and mixed-precision arithmetic are very attractive approach today, and this work demonstrates the possibility of applying them into resolving the most complicated scientific project such as ocean model. Firstly, in order to scale the ocean model onto the large-scale and extremely complicated supercomputer, four-level parallel framework are proposed. Sophisticated tuning techniques such as customizable domain decomposition according to the grid feature, are included as well. This enables the capability of fully utilizing the rich computing resources of the new system. The new feature of the system, thread-level RMA communication mechanism, is also wisely used for algorithms such as composite blocking, to further optimize the bandwidth performance. Moreover, mixed-precision optimization is proposed and performed on certain part of the algorithms. With sufficient material and proof to support its feasibility. Significant performance speedup is obtained thanks to these innovations. About 20 million cores are used for the large-scale test, and sustained performance of nearly 2 Petaflops. These innovations are solid, and can be very interesting to domain experts that expect to perform similar work by using the new Sunway supercomputer or other supercomputers with alike architecture. Besides, the work is also very useful for computer scientists like me, to rethink the architecture design for better supporting numerical scientific applications. I have no further comments, but some minor suggestions.

***Reply: Thank you very much for your recognition of our work, which is important for us. In fact, this work took more than one year, and we re-wrote almost all the code to port and then to improve the parallel efficiency. Fortunately, we achieved up to 99.29% parallel efficiency with a resolution of 500 m using 27,988,480 cores, which should be the largest***

*parallel scale on the ocean simulation up to now. We are happy it is beneficial to your work and the community.*

Minor issues:

1. What is the portability of proposed methods of this work? Eg, to other models, or other applications from different domain.

*Reply: Thank you. Several new optimization approaches proposed, such as a four-level parallel framework with longitude-latitude-depth decomposition, a multi-level mixed-precision optimization method that uses half-, single-, and double-precision, are the methods of general applicability. We test these optimization approaches in the NEMO, but these can be incorporated into other global/regional ocean general circulation models (e.g., MOM, POP, ROMS, etc.). Moreover, the optimizations on the stencil computation can be applied to any model with stencil computations.*

*The above description was added in the Conclusion and Discussion section in the revision.*

2. What is the lesson learned of this work, in terms of architecture design for future supercomputing systems.

*Reply: Thank you. From the view of future ocean simulations, we propose the following aspects that should be paid more attention to.*

*The first is the memory bandwidth. The architecture of the new generation of Sunway processors (SW26010 Pro) adopts a more advanced DDR4 compared with the original SW26010. It not only expands the capacity but also greatly improves the DMA bandwidth of the processor. In this work, we resolved the memory bandwidth problem through fine-grained data reuse technology, thus improving the memory bandwidth utilization rate to approximately 88.7% for DDR4 and paving the way for the ultrahigh scalability of NEMO. However, we noted that the efficiency increases using single precision instead of double precision. As the peak performance of SW26010 Pro are the same between double and single precision, the increased efficiency is mainly from the reduced memory access with*

*changing double precision to single precision. It indicates that the memory bandwidth is still a bottleneck.*

*The second is the half-precision. The finer resolution and more complex processes are the main directions of OGCMs development. Therefore, computational efficiency becomes more and more important. Reduced precision is an effective method for improving efficiency. In the past decade, ECMWF successfully implemented the single-precision in the weather forecast system, which achieves about 40% greater computational efficiency almost without degrading forecast quality. The savings in computational cost mainly come from reduced memory access. The half-precision can not only reduce memory access but also improve the floating-point computing power. Our results also prove that implementing the half-precision in the model can increase the computational efficiency, although we only revised several subroutines of NEMO. We noted that the new architectures of HPC become to support the half-precision, but the support is still incomplete, e.g., the transcendental function cannot be calculated with half-precision in the new generation Sunway.*

*The third is the I/O efficiency. The output data volume becomes larger with finer resolution. In our work, we tried to store the results with 1 km resolution, but the data volume is more than 65 TB per output, which took more than one-day of clock time. Therefore, the I/O efficiency is still a limitation for finer resolution models.*

3. What are the major obstacles that caused the performance loss. What can be done in future to further improve the performance of HPC ocean modeling, from perspectives of both model development and computer design.

*Reply: Thank you. We think the major performance losses are from the communications and bandwidth. From the view of software and hardware co-design, the following should be focused on in the future.*

*The first is the decomposing and load-balance. For the model design, we should find the proper decomposing scheme to fully utilize the computer architecture. Besides the time dimension, solving an ocean general circulation model is a 3-dimension problem, with longitude, latitude, and depth. Usually, only the longitude-latitude domain is decomposed. In our work, driven by the RMA technology, we achieved the longitude-latitude-depth*

*domain decomposed, which enables the better largescale scalability. Meanwhile, keeping a good load-balance is also important for scalability. For the computer design, the RMA technology is a good example, which enables the longitude-latitude-depth domain decomposing. In other words, the high communication bandwidth between different cores or nodes will help to largescale scalability.*

*The second is communications. With the increasing processes used for model simulation, the ratio of communications time to computational time will become higher. For the model design, the first thing is to avoid the global operator, such as ALLREDUCE, and BCAST, which will take more time with increasing the processes. Otherwise, it will be the crucial bottleneck. Meanwhile, we also should pack the exchanged data between different processes as much as possible. For the computer design, the low latency will help in saving the communications time.*

*The third is reduced-precision. The results of our work demonstrate that there is a great potential to save computational time by incorporating the mixed double-, single-, and half-precision into the model. For the model design, we should understand the minimum computational precision requirements essential for successful ocean simulations, and then revise or develop arithmetic. For the computer design, the support for half-precision should be considered in future.*

*Overall, the above are only several examples for further improving the performance of ocean modeling from perspectives of model development and computer design. Furthermore, other aspects such as I/O efficiency, and the trade-off between precision and energy consumption should also be considered. And it should be noted that these suggestions are from different aspects of the model and computer development and need to be considered based on the software and hardware co-design ideology.*

*The above description was added in the Conclusion and Discussion section in the revision.*

**§3 Others**

*1. We further polished the manuscript with minor changes about the grammar, and do not list the changes one-by-one here.*